# High throughput platform technology for rapid target identification in personalized phage therapy

Fereshteh Bayat [1], Arwa Hilal [1], Mathura Thirugnanasampanthar [2], Denise Tremblay[3,4,5], Carlos D. M. Filipe [2], Sylvain Moineau [3,4,5], Tohid F. Didar [1,6,7] ✉ & Zeinab Hosseinidoust [1,2,7,8] ✉

As bacteriophages continue to gain regulatory approval for personalized human therapy against antibiotic-resistant infections, there is a need for transformative technologies for rapid target identification through multiple, large, decentralized therapeutic phages biobanks. Here, we design a high throughput phage screening platform comprised of a portable library of individual shelf-stable, ready-to-use phages, in all-inclusive solid tablets. Each tablet encapsulates one phage along with luciferin and luciferase enzyme stabilized in a sugar matrix comprised of pullulan and trehalose capable of directly detecting phage-mediated adenosine triphosphate (ATP) release through ATP bioluminescence reaction upon bacterial cell burst. The tablet composition also enhances desiccation tolerance of all components, which should allow easier and cheaper international transportation of phages and as a result, increased accessibility to therapeutic phages. We demonstrate high throughput screening by identifying target phages for select multidrug-resistant clinical isolates of *Pseudomonas aeruginosa*, *Salmonella enterica*, *Escherichia coli*, and *Staphylococcus aureus* with targets identified within 30-120 min.

The spread of antimicrobial resistance (AMR) is considered a major global challenge, claiming more than 700,000 lives per year, with a projected increase to 10 million by 2050[1]. The looming global crisis of AMR and urgency to identify new therapeutics to fight against acute and chronic bacterial infections has garnered the attention of researchers, clinicians, and regulatory bodies towards bacteriophage therapy (i.e., the use of bacteriophages to fight bacterial infections). Bacteriophages, or phages for short, are viruses that exclusively infect bacteria in a highly targeted manner[2]. This targeted killing action is in stark contrast to the indiscriminate action of antibiotics and is a major

advantage of phages over antibiotics, promising minimal disruption to our microbiota[3–5], while simultaneously challenging decades of experience in development of antimicrobial therapies.

Phage therapy often relies on labor-intensive and time-consuming methods which may limit its potential. The design and administration of phage therapeutics require a fundamental shift in the way we think about antimicrobial therapies. Among others, there is a need for custom-designed tools and technologies that meet the standard of care in modern medicine. One of the challenges in the clinical practice of phage therapy is related to the targeted action of phages. Human

[1]School of Biomedical Engineering, McMaster University, Hamilton, Ontario, Canada. [2]Department of Chemical Engineering, McMaster University, Hamilton, Ontario, Canada. [3]Département de biochimie, de microbiologie et de bio-informatique, Faculté des sciences et de génie, Université Laval, Québec City, QC, Canada. [4]Groupe de recherche en écologie buccale, Faculté de médecine dentaire, Université Laval, Québec City, QC, Canada. [5]Félix d'Hérelle Reference Center for Bacterial Viruses, Université Laval, Québec City, QC, Canada. [6]Department of Mechanical Engineering, McMaster University, Hamilton, Ontario, Canada. [7]Michael DeGroote Institute for Infectious Disease Research, McMaster University, Hamilton, Ontario, Canada. [8]Farncombe Family Digestive Health Research Institute, McMaster University, Hamilton, Ontario, Canada. ✉e-mail: didar@mcmaster.ca; doust@mcmaster.ca

phage therapy is most effective when implemented as a personalized therapy, as demonstrated by an increasing number of clinical case reports[5–7]. Personalized phage therapy begins with the isolated pathogenic bacterial strain deemed to be resistant to available antibiotics and its screening against large libraries of therapeutic phages and/or libraries of environmental samples expected to contain phages

(Fig. 1a-i). These libraries can contain hundreds of phages and thus require susceptibility profiling technologies that can be implemented rapidly and in a high throughput format. The current gold standard in susceptibility testing, is the spot test[8], a culture-based method which involves a long incubation period of overnight to several days depending on the targeted bacteria (Fig. 1a-ii)[9]. Although widely used

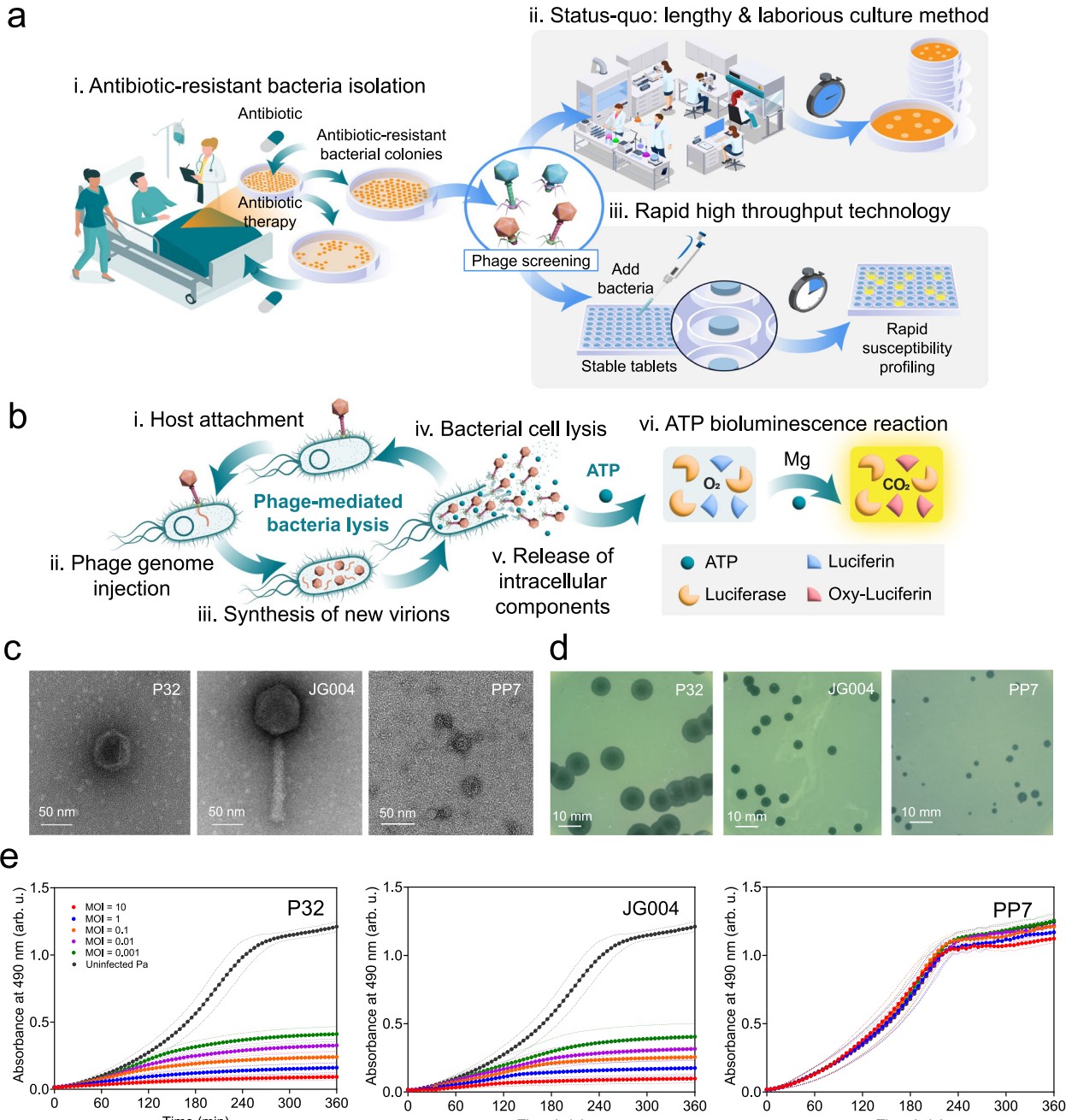

**Fig. 1 | Schematic of vision and phage characterization. a** Personalized phage therapy. The first step in personalized phage therapy is the isolation of treatment-resistant bacterial strain from patient (i), followed by employing slow culture-based methods to screen and select therapeutic phages (ii). We propose a single-tablet technology for rapid, high-throughput screening of therapeutic phages (iii). **b** Phage-mediated ATP release and detection. Phage-mediated lysis of bacterial cell starts when a phage virion encounters the host cell and attaches to specific phage receptors (i), this is followed by phage genome entry into the host bacteria (ii). This starts a cascade of events leading to hijacking of bacteria replication machinery for synthesis of new progeny phage (iii), which ultimately leads to host cell lysis (iv),

and release of progeny virions along with a burst of ATP (v). ATP reacts with luciferin to form luciferin adenylate, which is oxidized by the luciferase enzyme in the presence of magnesium to form oxyluciferin, $CO_2$ and adenosine monophosphate (AMP), which results in light emission (vi). **c** Transmission electron micrographs of three phages namely P32, JG004 and PP7 (two independent samples per phage) **d** Plaque formation on Pa lawns created by P32, JG004 and PP7. **e** Metabolic activity of uninfected and infected (MOI = 10, 1, 0.1, 0.01, 0.001) Pa cultures ($n = 3$, mean ± SD). All reported values are the mean of three biological replicates and associated error bars shown as dashed lines represent standard deviation from the mean. arb. u. stands for arbitrary unit. Source data are provided in Source Data file.

in research labs, the spot test method is laborious and slow. In the common spot tests, as a result of phage lysing bacterial cells and the release of progeny phages, clear zones are formed on solid medium[9]. Liquid assays are also performed to monitor bacterial lysis as a result of phage infection and they include tracking the bacteria culture turbidity (optical density) as a result of phage lysis[10]. The main hurdle of monitoring optical density of a bacteria culture is that bacteria cell debris can contribute to turbidity of the culture and affect the results negatively[11]. The overall metabolic activity of phage-infected bacterial cultures can also be monitored using various biochemical assays that have gained much attention recently and can be used to analyze properties related to phenotypes, namely cell growth and respiration[11]. However, these metabolic assays are detecting the phage-mediated bacterial cell lysis indirectly.

The second challenge is that a universal phage library is non-existent and patients, researchers, and clinicians pursuing phage therapy depend on limited libraries maintained by local and, in very few cases, national or military research labs[3]. Successful implementation of personalized phage therapy, therefore, requires streamlined communication and sharing between phage libraries at the national and international levels[12]. An effective solution is required to address both aforementioned challenges in personalized phage therapy, namely a reliable and rapid high throughput technology with shelf-stable and portable assay reagents that can be readily shipped around the world to be employed at the point of care with minimal infrastructure or training. To meet these design criteria, we re-imagined a therapeutic phage library in the physical format of stable and all-inclusive solid tablets, each tablet encapsulating a single phage along with the biochemistry to detect phage-mediated cell lysis (Fig. 1a-iii). Detecting the released adenosine triphosphate (ATP) as a proxy for phage-mediated bacterial lysis can be implemented to develop fast and high throughput phage library screening (Fig. 1b). To do so, two major barriers in the biochemistry of ATP detection should be overcome. ATP can be readily detected with firefly luciferase[13], which is a heat-labile enzyme and quickly deactivated at temperatures above 30 °C[14]. For human therapeutic applications, however, lytic activity of bacteriophages must be detected at 37 °C, which is the optimal temperature for the growth and metabolic activity of clinical bacterial pathogens and thus for phage lytic action. In addition, commercially available ATP detection assays require stagewise addition of reaction components, complicating high throughput implementation.

Here, we utilize materials technology to stabilize and encapsulate all reagents required for ATP detection along with each phage in a sugar-based matrix that prevents thermo-inactivation of the enzymes to overcome barriers in detection. Optimizing the biochemical reaction in the presence of this matrix enables a hassle-free, one-pot biochemical reaction, stabilized in the form of a solid tablet that preserves the activity of the enzymes at physiological temperatures as well as phage infectivity. The latter promises easier and cheaper transportation and as a result, easier screening of decentralized phage libraries and increased accessibility to phage therapy. We demonstrate the utility of this platform technology for high throughput implementation by phage susceptibility screening of selected isolates of *Pseudomonas aeruginosa* (Pa), *Salmonella enterica*, *Escherichia coli*, and *Staphylococcus aureus* against in-house phage libraries.

## Results and discussion
### Phage-mediated bacteria lysis
To develop a one-tablet assay based on the detection of phage-mediated lysis, we started by selecting three phages from our in-house phage library with different levels of bacteria lysis ability. Bacteriophages vB_Pae-Tbilisi32 (P32), JG004, and PP7 were propagated using our host bacterial strain, *P. aeruginosa* PAO1 (Pa). Transmission Electron micrographs of the three phages are shown in Fig. 1c. P32 is a podophage and has a very short tail[15], JG004 is a myophage[16] with an

isometric head and a contractile tail. Both of these phages have a double-stranded DNA genome. Phage PP7 belongs to *Leviviridae* family (single-stranded RNA genome) and has an icosahedral capsid with an approximate diameter of 30 nm (Fig. 1c)[17].

Figure 1d also plaques generated by each of the three phages on Pa bacterial lawn, indicative of the ability of these phages to successfully infect and lyse Pa[18,19]. Plaque morphology can sometimes provide qualitative yet important information regarding phage characteristics. For example, larger plaques may be indicative of a larger burst size (number of progeny phages released from the infection of a single bacterial cell)[20] and shorter latent period (period between phage adsorption and release of progeny virions)[21]. As seen in Fig. 1d, P32 generates the largest plaque, followed by JG004, and then PP7. To further characterize phage-mediated bacteria lysis, we generated kinetic kill curves which showed the change in metabolic activity of the bacterial culture challenged with the three phages, using the XTT colorimetric assay (Fig. 1d). Pa was infected at different MOIs (multiplicity of infection, defined as the ratio of infectious virions to bacterial cells in a culture)[22]. For all experiments, starting concentration of bacteria was kept constant (~$10^7$ CFU/mL). Both phages P32 and JG004 (Fig. 1d) significantly suppressed bacterial growth, as indicated by a low metabolic activity. The decrease in bacterial metabolic activity was slower at lower MOIs, as expected. It is noteworthy that although PP7 formed visible plaques and lysis zones on a solid medium, it did not suppress bacterial growth in our liquid assays (Fig. 1d). These trends agree with Pa kill curves based on optical density (Supplementary Fig. 1) and highlight a very important bias of standard methods in phage susceptibility testing, specifically as it pertains to phage application for therapy and biocontrol. Indeed, a clearing on a bacterial lawn (plaque or spot test) does not necessarily signal the ability of a phage to control the population of bacteria in a liquid culture, and as some studies have indicated, in in vivo models[23].

It is important to note here that phages may have at least three different cycles, namely lytic, lysogenic, and chronic (Supplementary Fig. 2). Through the lytic cycle (Supplementary Fig. 2a) a phage can lyse a bacterial cell in as short as ~20 min, leading to the release of the bacterial intracellular content in addition to tens or hundreds of progeny phages[24]. Regulatory agencies have historically only approved strictly lytic phages for human therapeutic use and environmental biocontrol because of concerns regarding horizontal gene transfer through the lysogenic cycle (Supplementary Fig. 2b, c)[25]. As our aim was to screen phage libraries for therapeutic potential, we focused our work on the detection of phages capable of bacterial lysis.

### Increasing signal-to-noise ratio and end-point detection of phage-mediated bacteria lysis
The next step towards realizing the high throughput, one-pot detection of phage-mediated bacteria lysis was to filter out the background signal. Phage stocks are obtained through the infection and lysis of bacterial cells leading to the release of progeny phages along with other intracellular components, including ATP. Therefore, we hypothesized that phage suspensions must be treated to reduce residual ATP molecules which can otherwise give rise to a strong background signal. The goal was to diminish the background bioluminescent signal at time zero, i.e., at the point of phage addition to the bacterial culture, which would in turn decrease the assay time and allow a signal to be discernable as soon as phage-mediated lysis of bacterial cells occurs. Supplementary Fig. 3a shows the background bioluminescence signal in phage suspensions at different stages of purification (including sterile filtration with 0.2 μm filters, PEG purification, supernatant, and ultra-filtration with 10 KDa and 3 KDa filters) and after diluting phages in fresh culture media. After PEG purification, the ATP concentration decreased significantly; however, ultrafiltration with 3 KDa filters failed to reduce ATP levels. As shown in Supplementary Fig. 3b, simply diluting concentrated phage suspensions in cell media was found to be

equally effective for reducing background ATP signal as the labor-intensive PEG purification technique. Based on these results, we used the dilution method to reduce the background bioluminescence signal and for preparing phage libraries. While the dilution method reduced the residual ATP signal, it did not completely eliminate the background bioluminescence signal. Thus, the background signal was deducted from the data for better visualization of the signal resulting from phage-induced ATP release.

For end-point detection of phage lysis, each phage was mixed with a culture of host bacteria at physiological temperature. Aliquots were collected periodically and added to ATP assay reagents in a stagewise manner at room temperature, before measuring the bioluminescence signal (Fig. 2a). Exponentially growing bacteria were infected with phage at MOIs of 10, 1, 0.1, 0.01, and 0.001. Figure 2b–f shows ATP release as a result of phage-mediated lysis, measured over a 3-h period. At MOI -10, bioluminescence signal was detected within 30 mins after the addition of phages P32 or JG004 to the bacterial culture (Fig. 2b). At lower MOIs, bioluminescence signal became detectable at later time points. The phage with the weakest lytic activity, PP7, showed a significant bioluminescence signal only at MOI = 10 and after 60–120 mins, while at lower MOIs, the signal was not significantly different from the uninfected Pa control, demonstrating the ability of the assay to differentiate strong and week lytic activity. Another trend to note is that for P32 and JG004 phages at MOIs of 0.01 and 0.001 (Fig. 2e, f), the signal intensity at the end of 3 h incubation period was visibly higher than the same time point at MOI = 10. This can be explained by the lower number of phages at the beginning of infection, which provides enough times for bacterial cells to grow and increase their population, Fig. 2g clearly shows that shortest time to detect a positive signal is with the highest MOI.

Another noteworthy trend is that even at very low phage to bacteria ratio of 1:1000 (MOI = 0.001), signal intensity from phage-infected samples is different from uninfected samples after 2 h of incubations (Fig. 2f). This likely shows the ability of the ATP detection biochemistry to discriminate between phage-mediated lysis and bacterial autolysis at very low MOIs. It is important to keep in mind that the concentration of phages curated in libraries is quantified in terms of plaque forming units, which reflects the number of infective particles capable of infecting the known host bacteria used to propagate the library phages at the stage of making the library. When screening for therapeutic phages against an unknown strain of bacteria (e.g., a multidrug resistant clinical or environmental isolate), the number of infective particles against an uncharacterized strain may be orders of magnitude lower than the original host strain, a concept known as efficiency of plaquing[26]. Therefore, demonstrating susceptibility profiling at low MOI (low efficacy of plaquing) is an additional advantage.

### One-pot biochemistry for detection of phage lytic activity
End-point detection of phage-mediated bacteria lysis involves periodic sampling of bacterial cultures, mixing with ATP reagents at room temperature, before measuring the bioluminescence signal. This technique is time-consuming and laborious, making it incompatible for high throughput screening applications. A one-pot format in which assay reagents are mixed with bacterial cultures at the start of phage infection cycle (Fig. 3a) would be a more desirable and practical approach when designing a high throughput screening technology. Figure 3b shows the bioluminescence kinetic curve for cultures infected with phage P32 or phage JG003 at MOI = 10 in a one-pot format. Notably, a detectable signal appears -30 mins post infection and peaks at 60 mins. Thereafter, the signal decays quickly until it becomes undetectable -4h post-infection. Signal detected from cultures infected with phage PP7, known to have weak lytic activity, was indistinguishable from uninfected cultures. The major shortcoming of the one-pot biochemistry of ATP detection, however, was that the bioluminescence signal produced under these conditions started to decay

after 1 h (Fig. 3b). The signal decay is likely the result of thermal inactivation of luciferin and luciferase at 37 °C.

Despite the temperature inactivation of luciferase, our data show that the one-pot format not only discriminates between uninfected and infected cultures but can also differentiate between strong (P32), moderate (JG004), and weak (PP7) lytic activity of different phages even at initial lower phage titers (lower MOIs) (Fig. 3d). This capability to resolve lytic activity makes our method as efficient as the gold standard plaque assay, while offering the obvious advantage of being amenable to high throughout format. A re-examination of Fig. 1d showed that the plaque morphology for each phage, which shows the combined effect of phage burst size and latent period and is thus a good indicator (although not the only indicator) for phage biocontrol efficacy, agrees with the trends observed in Fig. 3b, d. Latent periods and burst size of phages are among the characteristics used to evaluate phage virulence[27], but they may not be indicative of therapeutic potential in vivo as various conditions will affect the efficacy of phage therapy as mentioned by other researchers[28]. Phage P32 is reported to have a 20-min latent period and a burst size of 210 progeny phages on *P. aeruginosa*[15]. JG004 is reported to have a 31-min latent period and a burst size of 13 progeny phages on *P. aeruginosa*[16]. Latent period and burst size numbers may vary depending on host bacterial strain, stage of bacterial growth, and nutrient source; however, the general trend should hold. Figure 3b shows 25-30 mins are required from the time of phage addition to the time to detect the bioluminescence signal, which matches closely with the latent period of phages P32 and JG004[20]. As seen in Fig. 3c, the time to peak signal decreases as the concentration of phage increases (higher MOIs) as expected. The fact that phage P32 generated a stronger peak signal as compared to phage JG004 showed that more bacterial cells were lysed by P32 compared to JG004 (Fig. 3d). The peak signal correlates with the number of bacterial cells lysed. Having a large burst size can also affect the peak signal by having more progeny phages released to the surrounding medium, which can, in turn, infect and lyse more bacterial cells. Taken altogether, these findings confirm the feasibility of the one-pot chemistry for the detection of phage-mediated cell lysis and thus showing promise to form the basis of a high throughput susceptibility profiling technology.

### Optimizing the detection of phage-mediated lysis in a stabilizing sugar polymer matrix
The next step towards a reliable susceptibility assay was to address the rapid enzyme deactivation at physiological temperatures. To address this challenge, we selected pullulan, based on previously reported thermo-protective effects towards biomolecules[29,30], and trehalose based on reported desiccation protection towards viruses[31,32]. Pullulan and trehalose were added to a mixture of phage suspension and lyophilized ATP detection reagents and homogenized before the addition of bacteria (Fig. 4a). As shown in Fig. 4d, a discernable signal appeared 25–30 mins after all assay components were mixed with the bacterial suspension. The time to signal detection in the presence of the sugar mixture was comparable to the sugar-free mixture (Fig. 3b, c). This confirmed that the addition of sugar polymers did not interfere with the function of assay reagents or with phage infectivity. Moreover, the addition of sugars (pullulan-trehalose) did not significantly affect the peak bioluminescence signal, whereas the signal at 6 h post infection in cultures infected with either phages P32 (Fig. 4b) or JG004 (Fig. 4c) was -99% and 95% higher, respectively, than the negative control. In summary, the addition of pullulan-trehalose to the mixture stabilized the assay signal and significantly reduced signal decay. To further assess how closely the RLU signal follows the phage-mediated cell lysis, we designed an experiment with ATP standard solutions. ATP reaction solutions were prepared in the absence or presence of the sugar polymer mixture. ATP standard solutions with three concentrations (0.4, 0.01, 0.001 μM) were prepared in water. The RLU values correlated with 0.4 μM ATP standard solution was chosen at least 10 times

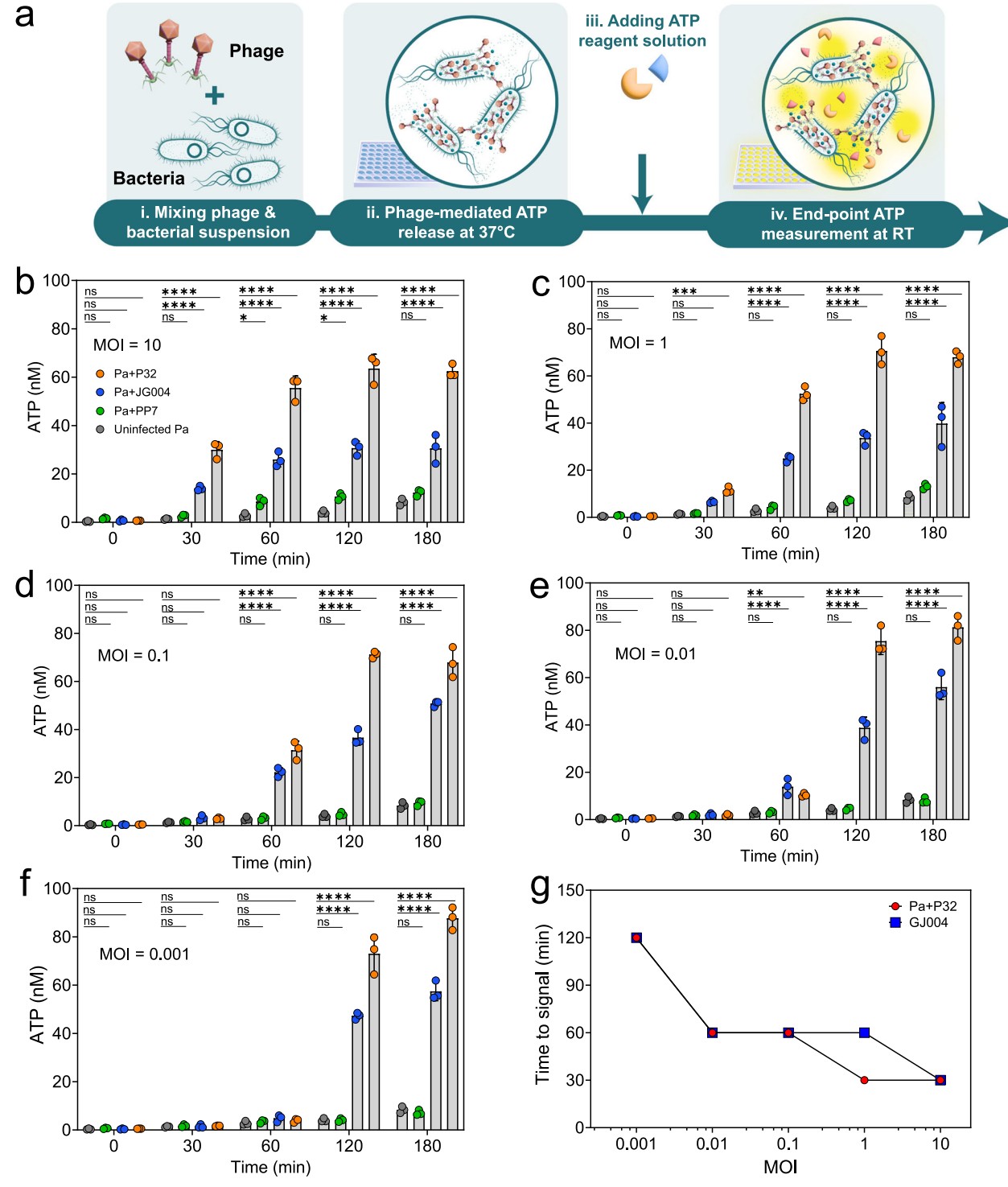

**Fig. 2 | Sequential assay for detection of phage-mediated ATP release.**
**a** Workflow schematic: Bacterial cultures were infected with phages at different concentrations (i) and incubated at 37 °C (ii), at different time intervals, ATP was measured at room temperature by adding ATP reagent solution which contains luciferin and luciferase as main components (iii), and bioluminescence signal was measured (iv). Concentration of ATP was calculated for bioluminescence signal measured from Pa individually infected with P32, JG004, and PP7 at different MOIs:

**b** MOI = 10, **c** MOI = 1, **d** MOI = 0.1, **e** MOI = 0.01, **f** MOI = 0.001. **g** time to signal for Pa infected with P32 and JG004 at different MOIs. All reported values for panels **b**–**g** are the mean of three independent experiments with $n = 3$ and associated error bars represent standard deviation from the mean. Statistical significance in panels **b**, **c**, **d**, **e** and **f** is derived from Two-way analysis of variance (ANOVA) and significance levels include *$P < 0.05$, **$P < 0.01$, ***$P < 0.001$, and ****$P < 0.0001$. Source data are provided in Source Data file.

larger than the amount of ATP released in the solution for the phage-mediated ATP detection in our one-pot assays. 0.01 and 0.001 μM ATP standard solutions were tested to see if at lower ATP values we are still able to detect consistent values in real-time phage-mediated RLU

signal monitoring at 37 °C for weak lysis activity. The activity of the ATP reagent solution with and without sugar polymers were also tested in four conditions including no previous exposure at 37 °C (tested right after preparation at room temperature), one, three and 6 h

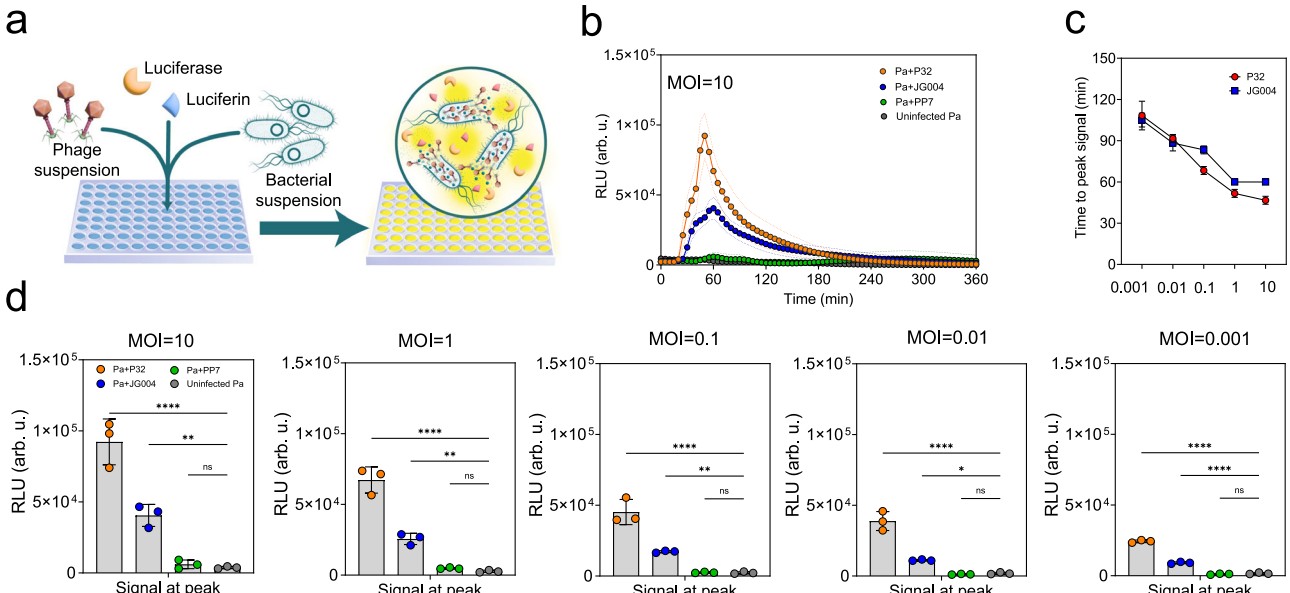

**Fig. 3 | One-pot ATP bioluminescence assay. a** Workflow schematic. ATP one-pot assay includes adding all ATP bioluminescence assay reagents (luciferin and luciferin as major components) and phage, followed by adding bacterial suspension and incubating the plate at 37 °C for real-time measurement of phage-induced ATP release. **b** Kinetic ATP measurement for Pa infected with P32, JG004, and PP7 at MOI = 10 and uninfected Pa. (the dashed line shows standard deviation from the mean for three independent experiments). **c** Time to peak signal for P32 and JG004 at different MOIs (values are the mean of three independent experiments, and associated error bars represent standard deviation from the mean). **d** Signal at peak for Pa cultures infected with P32, JG004, and PP7 at different MOIs including MOI = 10, MOI = 1, MOI = 0.1, MOI = 0.01, MOI = 0.001. All reported values are the mean of three independent experiment with $n = 3$, and associated error bars represent standard deviation from the mean. Statistical significance in **d** is derived from one-way analysis of variance (ANOVA) and significance levels include *$P < 0.05$, **$P < 0.01$, ***$P < 0.001$, and ****$P < 0.0001$. Source data are provided in Source Data file.

incubation at 37 °C. ATP standard solutions were added to the above-mentioned treated and untreated ATP reactions in the presence and absence of the sugar mixture, and the signal was measured at time zero, followed by continuous measurements for up to 3 h. The results confirmed that sugar polymer was able to stabilize the ATP bioluminescence assay at 37 °C, and the loss of signal was not significant after exposure at 37 °C in all three ATP standard concentrations. On the other hand, in the absence of sugars, the ATP reagent solution led to a significant decrease in RLU signal after incubation at 37 °C (Supplementary Fig. 4). In addition, the kinetic RLU signal monitoring showed that the rate of signal loss was also higher when no sugar was present in the ATP reagent solution (Supplementary Fig. 4). Specifically, the loss of signal after 6 h incubation at 37 °C resulted in an ~90% signal loss (Supplementary Fig. 5). This confirms that ATP bioluminescence assay in the presence of sugar polymers has been able to detect the ATP release as a result phage-mediated bacterial cell lysis.

We further demonstrated that the sugar mixture can be dried and cast into a tablet format, encasing all the assay components into a sugar matrix. As shown in Supplementary Fig. 6, titer loss for phage encased in the sugar polymer matrix was less than one log for all three phages after a 30-day storage under ambient conditions. In the absence of the sugar polymer matrix, phages showed weak desiccation tolerance, with PP7 being the least stable one with a 3-log reduction after 30 days. We then evaluated the stability of all assay reagents in a dehydrated tablet form. ATP assay components were first mixed with pullulan-trehalose and then phage suspensions in a well plate and dried under nitrogen airflow. Following a week-long storage under vacuum, dried tablets were reconstituted. As shown in Fig. 4e, the rate of bioluminescence signal from the reconstituted tablet assay was slower compared to the liquid assays. Although the signal intensity was lower compared to the original liquid assay, the RLU signal of the phage-containing tablets compared to control uninfected Pa was visibly higher showing a successful detection of the phage-mediated bacterial cell lysis. The signal intensity depends on the dissolution rate

of the tablets in water, which in turn controls the release of phages and ATP reagents into the mixture. We believe that the slower increase of the bioluminescence signal from the reconstituted tablet is a small trade-off for the portability, improved storage, and ease of use afforded by the solid tablet format (Fig. 4d). The all-inclusive tablets also preserved their stability after 4 weeks of storage at room temperature under vacuum (Supplementary Fig. 7).

Lastly, it should be noted that the drying of the ATP reagents and phages in the absence of sugar polymers led to a complete loss of signal after the 1-week storage period, as shown in Fig. 4f, g. This clearly shows the importance of sugar polymers in preserving phages and ATP assay components, which is particularly advantageous for ease of transportation and point-of-use accessibility, including disaster scenarios as well as underdeveloped and remote regions.

### Screening different bacterial species against phage libraries using sugar-based ATP bioluminescence assay

We have selected four different bacterial species, namely *P. aeruginosa*, *S. enterica*, *E. coli*, and *S. aureus* with two strains per species to assess the feasibility of the ATP bioluminescence assay in identifying target phages against bacterial strain of interest. The information regarding all the strains is included in Supplementary Table 1.

We randomly selected two multidrug resistant clinical isolates of *P. aeruginosa* from an in-house library of isolates from Hamilton Health Sciences. The antibiotic resistance profile for these strains is shown in Supplementary Table 1[33]. The two strains were given the designations C0072 and C0335 and were isolated from patients with urinary tract infection and arm wound infection respectively. We conducted phage susceptibility screening against our in-house phage library containing seventeen phages, with mixed host strains to control for false positive signals, using 1-week-old, all-inclusive tablets stored under ambient conditions. Figure 5a shows the general workflow for these assays. The kinetic RLU signal monitoring results of the tablet-based, one-pot assay shows a significant, but slow rise in bioluminescence signal after

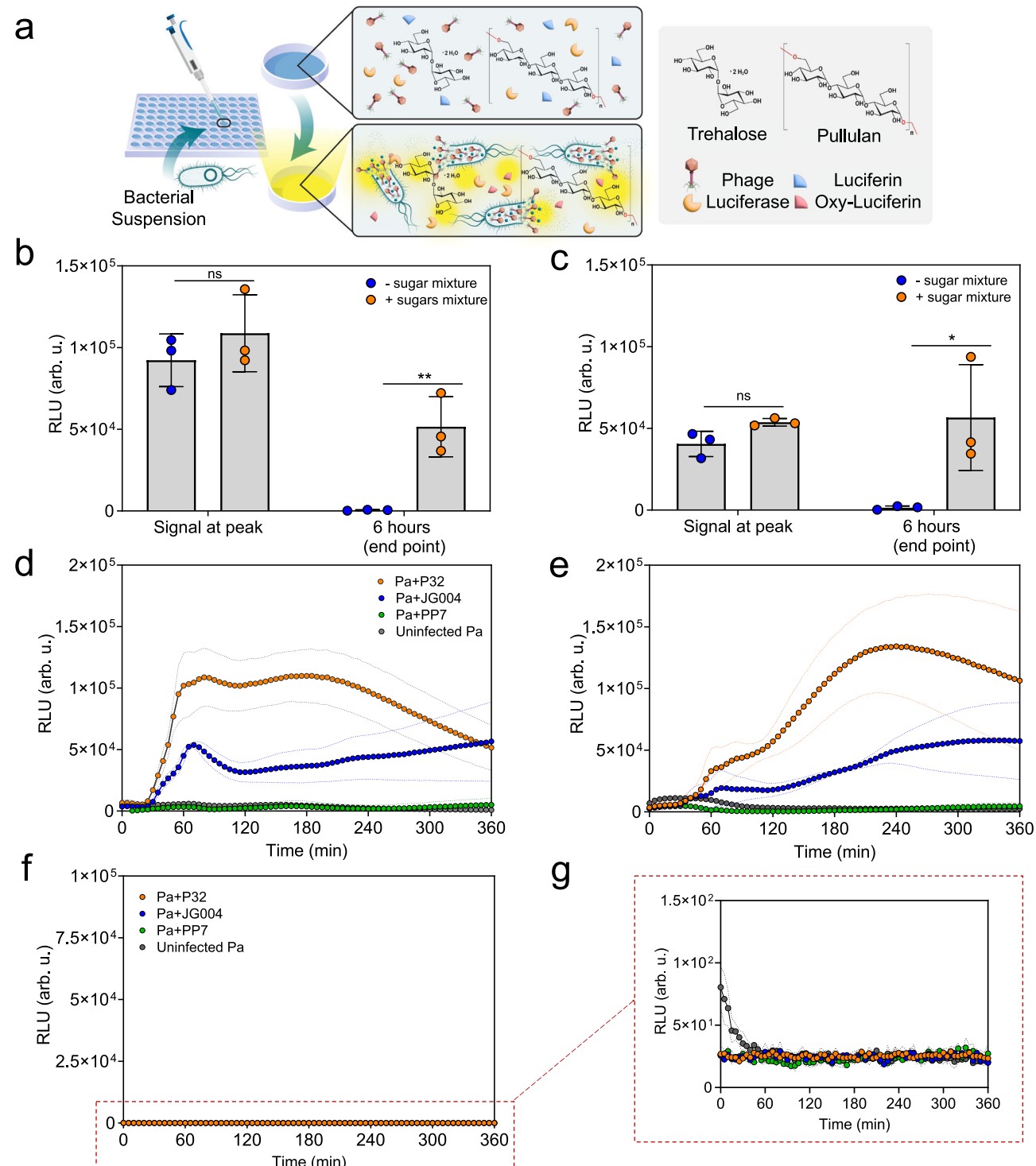

**Fig. 4 | One-pot ATP assay in the presence of stabilizing sugar polymer matrix.**
**a** Workflow schematic. Bacterial suspensions are added to 96-well plate containing phages mixed with ATP reagents in sugar mixture. **b** Bioluminescence signal at peak for Pa cultures infected with phages in the presence versus absence of sugar mixture in fresh liquid one-pot ATP assay. **c** Bioluminescence signal at peak and assay end point of 6 h for JG004 infected Pa cultures in the presence versus absence of sugar mixture in fresh liquid one-pot ATP assay JG004. Error bars in **b** and **c** graphs show the statistical analysis based on unpaired *t*-test, associated error bars represent standard deviation from the mean of three independent experiments (*n* = 3). Significance levels include *$P < 0.05$ and **$P < 0.01$. **d** Continuous measurement of

bioluminescence signal for Pa infected with P32, JG004, or PP7 in fresh liquid sugar mix solutions. **e** Kinetic measurement of bioluminescence signal for Pa infected with P32, JG004, and PP7 in reconstituted 1-week old tablets. **f** Kinetic measurement of bioluminescence signal for Pa infected with P32, JG004, and PP7 in rehydrated phage and enzymes dried in the absence of sugar polymers showing complete loss of signal after 1-week storage. **g** Same data as **f** but with a narrower *y*-axis scale, clearly showing the loss of signal. Dashed line in **d**, **e**, and **g** show standard deviation from the mean of three independent experiments (*n* = 3). Source data are provided in Source Data file.

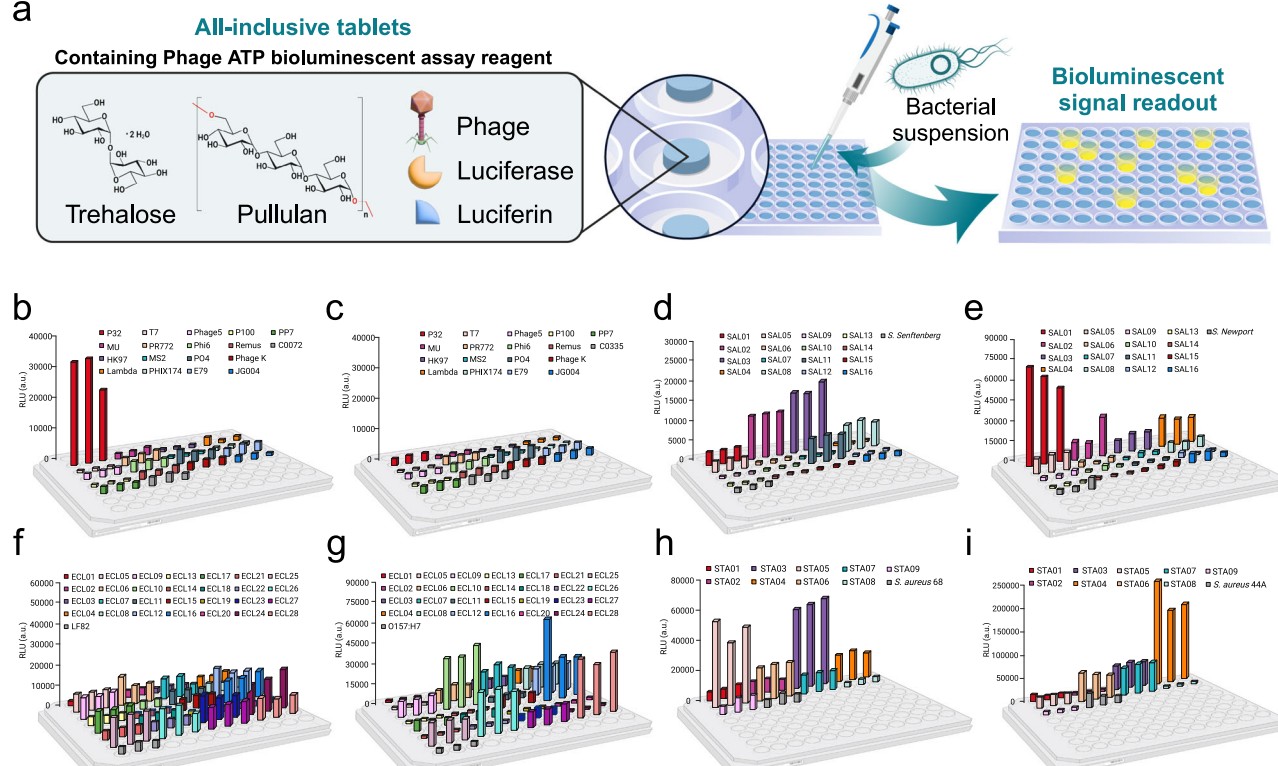

**Fig. 5 | Large-scale phage library screening of stabilized phage biobank.**
**a** Proposed workflow for screening a bacterial isolate against a phage library using our platform technology, where the phage library is screened by detecting phage-mediated ATP release using ATP bioluminescence assay in the presence of sugar polymers stabilizers. The assay can be conducted with fresh liquid sugar-polymer based ATP reagent solution or conducted on reconstituted all-inclusive sugar-based tablets. **b** The urinary tract infection isolate, *P. aeruginosa* C0072 and **c** the arm infection isolate, *P. aeruginosa* C0335 were screened against an in-house library of all-inclusive tablets. Bioluminescence signal was recorded every 5 min and the snapshot of the signal at time = 60 min is shown for each well. **d** the human blood isolate, *S. enterica* serovar Senftenberg were screened against the same *Salmonella* phages in the presence of sugar polymer stabilizers. The presented data are the snapshots at 90 min time point and **e** The sewage isolate, *S. enterica* serovar Newport screened against a library of 16 Salmonella phages (listed in Supplementary table 2). The present data show RLU signal at time = 120. **f** The Crohn's disease isolate, *E. coli LF82* were screened against a library of *E. coli* phages containing 28 *E. coli* phages listed in Supplementary table 3 in the presence of sugar polymer stabilizers. The presented data are the snapshot at 100 min time point for both *E. coli* strains and **g** The fecal isolate, *E. coli* O157: H7. **h** The snapshot of data at time = 170 min for *S. aureus* 68 strain and **i** the snapshot at time=100 min for *S. aureus* 44 A strain screened against a library of *S. aureus* phages containing 9 phages (listed in supplementary table 4). Data points represent 3 replicates. Panels b-i Created with BioRender.com released under a Creative Commons Attribution-NonCommercial-NoDerivs 4.0 International license. Source data are provided in Source Data file.

30 min for *P. aeruginosa* C0072 infected with P32 (Supplementary Fig. 8a). As snapshot of the signals from the microtiter plate is illustrated in Fig. 5b after 90 min, showing strong signals form the identified phages. As shown in Supplementary Fig. 8b, for the clinical isolate C0335, the bioluminescence assay did not show any rise in signal with any of the phages in the library (also demonstrated in Fig. 5c). However, a faint signal was observed for one of the phages (JG004) with a delay. Phage susceptibility was confirmed with optical density assay, and a spot test (representative images shown in Supplementary Fig. 8a, b) as well as XTT metabolic activity monitoring assay (Supplementary Fig. 9). The clinical isolate C0072 showed an obvious clearance with P32 (Supplementary Fig. 8) but clinical isolate C0335 isolate did not show susceptibility to any of the phages in the library, although a very faint clearing was observed with JG004 (Supplementary Fig. 8, spot test), which may correspond to the faint signal after 180 min (Supplementary Fig. 10). The identified phage for C0335 is clearly very weak and thus not recommended for phage therapy/biocontrol applications. We calculated the efficiency of plaquing of the identified phage on C0072 to be 0.77, confirming a high infection efficiency[34]. These data demonstrate that tablets, stored under ambient conditions, could identify phages capable of infecting clinical bacterial isolates. These data illustrate that the all-inclusive solid tablet format is capable of rapid screening MDR isolates against phage

libraries with high signal to noise ratio and reliability comparable to gold standard culture techniques, as well as metabolic assays.

To further challenge our platform technology, we screened two different strains of *E. coli* (O157: H7 and LF82) against a library of 28 *E. coli* phages, two strains of *S. enterica* (serovar Newport and serovar Senftenberg) against a library of 16 *Salmonella* phages, and two strains of *S. aureus* against a library of nine *S. aureus* phages from the Felix D'Herelle Reference Center for Bacterial Viruses, in the presence of sugar polymers. A full list of coded phages present in these two libraries can be found in Supplementary Tables 2–4. The received phages had titers of $10^8 – 10^{10}$ PFU/mL and were used without dilution to closely mimic a real life scenario where our technology is proposed to be used, when the initial titers in the library are not known for the bacteria of interest. Figure 5d, e shows a snapshot of the signals from the salmonella phage library after 90 to 120 min, with signals form identified targets clearly higher than the background signal. Examining the kinetic bioluminescence curves (Supplementary Fig. 11) show that the signal to noise ratio is starting to decrease at 90 min, clearly marking the phages that lead to lysis of the bacterial cell and release of ATP, which become clear at 120 min. The same trend can be observed for the *E. coli* phage library, where the kinetic bioluminescence curves (Supplementary Fig. 12) show that phages causing lysis of the bacterial cell and release of ATP are identified with a high signal to noise ratio

within 100 min (3D snapshot presented in Fig. 5f, g). In the *S. aureus* phage library, five phages were identified withing 170 min against the *S. aureus* 68 strain (Fig. 5h). However, a wide range of response times observed when screening this strain against our phage library as evident in the kinetic one-pot ATP curves (Supplementary Fig. 13a). Among the five target phages against *S. aureus* 68 strain, STA03 and STA06 showed ATP signal within 30 min and STA04 within 60 min. On the other hand, STA05, and STA07, both with very weak spot test signal, showed bioluminescence signal at later time within 130 min and 170 min, respectively. For *S. aureus* 44A, all the target phages were identified within 100 min (Fig. 5i). The kinetic one-pot ATP curves for *S. aureus* 44 A is shown in Supplementary Fig. 13b.

In addition, we benchmarked our method against gold standard culture methods, namely spot tests on semi solid media (end point detection) and kill curves (monitoring culture optical density at 600 nm, OD600). These data are presented in Supplementary Figs. 11–13 and show strong agreement with culture methods, verifying the reliability of our platform. Examining the data summarized for comparison with bioluminescence assay in Supplementary Tables 5–7 further reveals the utility of ATP detection as a direct measure of bacterial lysis, as opposed to indirect measures such as kill curves, specifically that certain phages that showed a strong signal with kill curves, showed no ATP signal, and no or very faint spot test clearing. What gives the ATP detection method an edge over other components is that it relieves a major bias shared by optical density and metabolic activity monitoring, a notable example being detection of slowed growth not accompanied by bacterial lysis (which happens for chronic phage or in some cases of lysogeny) versus bacterial cell lysis and destruction. The burst of ATP detected in our method can only happen if the bacterial cell is compromised and thus the detection of phage-mediated ATP release directly detects cell lysis, while other methods measure infectivity indirectly by monitoring the phenotypic features of bacterial cultures. Moreover, ATP release shows a real time measurement of bacterial cell lysis which can provide information on phage dormant period on a particular bacterial strain.

It is important to note that certain phenomena may result in effects that could be mistaken for the effects resulting from the completion of the lytic cycle. For example, at high MOIs (usually >100 phages per bacterium), some phages can bind to the bacterial strain and lead to cell death through abortive infection without releasing progeny phages, a phenomenon known as "lysis-from-without"[35,36]. In addition to lysis caused by multiple phage absorption to the surface of bacterial cells, there are reports of abortive phage infection due to bacterial defense mechanism, which is not necessarily as a result of high MOI. Abortive infections can prevent phage propagation by putting bacteria in a dormant state or lead to their death, or mutual destruction of phage and bacteria[37]. Such effects can result in a zone of clearing in a spot test assay that would be indiscernible form the zone of clearing caused by the completion of the lytic cycle. However, when followed up by downstream quality control with serial dilutions, the same phage would fail to lead any plaques. In a kill curve assay that relies on turbidity monitoring, phage-related bacterial cell dormancy can show a decrease in turbidity that can easily be mistaken for the effect of phage lysis. While methods of direct lysis monitoring, such as ATP detection, have advantages over indirect methods (e.g., spot test and kill curves), they may also be affected by effects of complex phenomenon known to exist in bacteria-phage interaction, such as lysis-from-without or abortive infection. Thus, we would like to emphasize that our method does not replace a rigorous downstream quality control on the shortlisted phage identified though our rapid screening method, or any other method of choice. Furthermore, while the true MOI cannot be determined a priori during phage screening, theory suggests using a higher bacterial concentration may help avoid lysis-from-without. To investigate whether cell death without progeny phage production can be a source of bias in the phage screening

process, we expanded the range of MOI in our investigation. We infected three P. aeruginosa strains with high titers of phage JG004 at a wide range of MOIs (-10,000, 1000, 100, 1, 0.1, and 0.01). As shown in Supplementary Fig. 14, only the Pa strain was sensitive to phage JG004, and the other three strains did not show any clearance on bacterial lawns (spot test) or a decrease in turbidity in standard kill curves, or any increase in RLU signal in ATP bioluminescence assay at any of the MOI's tested (Supplementary Fig. 14b, c, d). As shown in Supplementary Fig. 14a, in the kill curves, turbidity increased at the beginning of the assay and peak turbidity decreased with increase in MOI, which could be partially attributed to lower number of uninfected bacterial cells at high phage concentration and partially to lysis from without. For the phage sensitive Pa strain, the kill curves at different MOIs barely exhibit any difference, with the growth being effectively hindered at every MOI and the curve being visually indiscernible at MOI's ≥ 1.

The signal from ATP bioluminescence assays, however, showed very interesting trends, highlighting the power of direct monitoring of phage-mediated bacterial lysis (as opposed to indirect monitoring with turbidity assays). The Pa strain infected with an extremely high MOI of -10,000, showed a significantly lower ATP bioluminescence signal compared to lower MOIs. The peak signal increased by lowering the MOI in the range of 10,000 to 10. However, the peak signal started to diminish by lowering the MOI further, while showing larger signal at the endpoint of the assay. It is noteworthy that lysis from without usually happens earlier than normal time of lysis (< 5 min), but it can also happen at normal lysis times[35], which is in line with our observation. The time to signal that we obtained for MOIs ≥ 10 was around 30 mins and increased to 60, 70 and 90 mins for MOIs 1, 0.1, and 0.01, respectively.

The observed trend in bioluminescence signal (Supplementary Fig. 14) can be explained based on the dynamic of phage-bacteria interactions. When phage concentration is extremely high, some abortive phage infection may happen due to lysis from without. It has been reported in the literature that at high MOIs, lysis from without can lead to reduced phage counts and cell death[35]. This may be the reason behind the observed decrease in bioluminescence signal at MOI > 10. At MOI < 1 the signal at the endpoint was larger as the phage population was smaller and some uninfected bacteria (theoretically 37% of the population)[22] could continue growing, which would lead to a higher signal at the end for MOI = 0.01.

The proposed method is an all-in-one solution that is amenable to high throughput implementation, automation, and stable to store and shipping with small footprint and without the need for a cold chain. We envision phage libraries stabilized and stored in 384-well microtiter plates along with the biochemistry of cell lysis detection, in redundancy. When needed, these microtiter plates will be shipped to the point-of-use, where the only task needed will be to rehydrate and add the target bacteria; both steps are automation-friendly and take seconds to minutes (not accounting for bacterial growth of the initial bacterial culture, which is the same for all other methods methods). This is all in contrast to conventional culture-based liquid assays (kill curves), other liquid assays (e.g. XTT assay), and semi-solid assays (spot test) for which the phage biobank is seldom replicated and shipped to point-of need, for susceptibility screening. Because some phages may be unstable during shipping, hosts (pathogenic) strains may also need to be shipped. At the point of screening, phages will have to be retrieved from storage (in a freezer, fridge or freeze-dried) one-by-one, either added to liquid bacterial culture (tube, microtiter plate) along with the host bacteria and any other reagents that may be needed, such as redox-sensitive dyes in case of an XTT assay, or divalent cations in any phage susceptibility assay of choice. In case of a spot test, the solid medium will have to be prepared in advance, which will add to the laborious preparation process. Even an ATP assay would be cumbersome and lengthy without the proposed tablets, since all reagents will have to be freshly added in a stage-wise fashion, and the

assay will have to be run at room temperature, which would significantly slow down the growth of pathogenic bacteria and phage-mediated lysis, while increasing assay time. The phage stocks stored in conventional format may lose titer from one to multiple logs; so, it is standard practice to check titer prior to screening, which will add to the required time. Supplementary Fig. 15 illustrates the overall screening time needed for our proposed method in comparison with the spot test, turbidity measurements, for a modest size library of ~100 phages and one person dedicated to the task. The turnaround time for the all-inclusive tablets could reduce to ~2h, while other methods usually require at least a few days. By having shelf-stable phage tablets, the only step required is to reconstitute the dried tablets, add bacteria and monitor the bioluminescence signal to directly detect phage-mediated cell lysis through ATP release.

Personalized phage therapy is becoming an alternative strategy for many patients suffering from infections resistant to all known antibiotics. However, access to phage therapy is challenged by the lack of a universal phage library. There have been efforts in curating phage libraries around the world, but due to the remarkable phage diversity, and the numerous challenges associated with maintaining such bio-banks, the more realistic forecast is that we may always need to screen multiple phages located in decentralized libraries find suitable phage(s) against a resistant infection. This is challenged also by the lack of rapid screening methods, which are laborious, time-consuming, with a slow response time. Here, we addressed both challenges and presented a path towards an improved routine practice of personalized phage therapy worldwide.

We re-imagined phage libraries not in the form of liquid lysates in a fridge, or even frozen stocks in a deep freezer, but as solid tablets packed in microtiter plates, stored on a shelf, ready for rapid high throughput screening with a plate reader when needed, and ready to ship with a moment's notice. Each tablet contains a phage stock along with enzymes and ions needed to detect the burst of ATP release during phage-mediated bacteria lysis, all stabilized in a sugar polymer matrix that protects the enzymes against degradation at physiological temperatures. The matrix also offers desiccation protection to phages and enzymes, making the tablets shelf stable and easy to store and to ship. In addition, the solid, all-inclusive tablet format eliminates the need for stage-wise addition of reagents, enabling high throughput screening.

The proposed approach offers certain advantages over the culture techniques for phage susceptibility profiling, namely a faster response time and compatibility with high throughput implementation, as well as environmental stability, eliminating the need for a cold chain or special packaging. It also offers a point-of-use implementation with minimal infrastructure and training, which will be particularly impactful in remote regions. Our technology goes beyond binary susceptibility profiling and can also semi-quantitatively determined the phage ability for biocontrol. With personalized phage therapy being increasingly practiced for treating antibiotic-resistant infections, these proposed all-inclusive phage-containing tablets will speed up the screening process to identify phages that target the pathogenic bacterial strain of interest.

## Methods
### Materials
Pullulan (PI20 food grade, 200 kDa) from Hayashibara Co., Ltd., Okayama, Japan was kindly provided by Dr. Carlos Filipe, Department of Chemical Engineering, McMaster University. D-(+)-Trehalose dehydrate, and XTT tetrazolium sodium salt, menadione, and acetone were purchased from Sigma Aldrich. Phosphate-buffered saline (PBS) tablets were purchased from VWR (Mississauga, ON, CA). Luria Broth (LB) and Tryptic Soy Broth (TSB) were purchased from Fisher Scientific (ON, CA). ATP bioluminescence Assay Kit CLS II was purchased from Millipore Sigma (Sigma-Aldrich, Oakville, ON). vB_Pae-Tbilisi32 (P32) and JG004 phages, and PAO1 (Pa) strain were purchased from DSMZ

(Germany), and PP7 phage and *E. coli* 157:H7 bacterial strain from Université Laval (QC, Canada). LF82 was generously provided by Dr. Brian Coombes (Department of Biochemistry & Biomedical Science, McMaster University). Two clinical *P. aeruginosa* strains including C0072 and C0335 were obtained from IIDR database at McMaster University (Please refer to Supplementary Table 1 for additional information). *Salmonella* strains along with 16 *salmonella* phages, 28 *E. coli* phages, and two *S. aureus* strains along with 9 *S. aureus* phages were obtained from the Felix d'Herelle Reference Center for Bacterial Viruses at Universite Laval (https://www.phage.ulaval.ca/en/home/). The details of bacterial species and phages can be found in the supplementary Tables 1, 2, 3 and 4.

### Bacterial culture and phage propagation
All frozen bacterial stocks were stored at −80 °C in 25% v/v glycerol. Overnight bacterial cultures were prepared by inoculating 3 mL of bacterial media with glycerol stock. *P. aeruginosa* and *E. coli* strains were cultured in LB media, while *S. enterica* and *S. aureus* strains were grown in TSB media. The inoculated media was incubated at 37 °C and 180 rpm for 16–18 h to promote bacterial growth. Overnight cultures were subsequently diluted 1:100 in 50 mL of fresh bacterial cell media and incubated for 2–3 h to allow bacterial cells to reach the mid-exponential growth phase. 10 μL of stock phage solution was introduced to bacterial cells at mid-exponential phase and the cultures were incubated for a further 6 h to allow for phage lysis of bacterial cells. The lysate was centrifuged for 20 min at 7000 rcf, and the supernatant was sterilized using a 0.2-μm-pore-sized filter and stored at 4 °C. For *E. coli* and *Salmonella* phages propagation, overnight bacterial cultures were grown in 10 mL of TSB from glycerol stocks at 37 °C with agitation. Bacterial overnight culture was diluted 1:100 in 10 mL of TSB and a scratch of phages from glycerol stocks was added. The mixture was incubated at 37 °C with agitation until lysis (~6h). Lysates were filtered through 0.45 μm filters.

Phage concentration (number of plaque forming units per milliliter, PFU/mL) was determined using the overlay technique[38]. Phage stock was serially diluted in LB or TSB media and plated on bacterial lawns to enumerate the number of infectious virions present in the stock.

### Efficiency of plating (EOP)
Efficiency of plating (EOP) measures the titer of phage infecting a non-host bacterial strain. EOP assay was conducted with P32 (propagated on host strain, Pa) for *P. aeruginosa* C0072 strain. Phage overlay assay was conducted in triplicates to obtain the PFUs on the test strain (C0072) and the host strain (Pa). EOP was calculated using the following formula[34].

$$EOP : \frac{\text{Average PFU on the test strain}}{\text{Average PFU on the host bacterium}}$$

Infection efficiency:
EOP $\geq 0.5$ (high production efficiency)
$0.1 \leq$ EOP $< 0.5$ (moderate production efficiency)
$0.001 <$ EOP $< 0.1$ (Low production efficiency)
EOP $\leq 0.001$ (inefficient)

### Phage purification
The aqueous two-phase method was used to purify filter-sterilized phage suspension[39]. Briefly, sterile 20 (w/v)% Poly(ethylene glycol) containing 2.5 M NaCl solution was added to the phage stock suspension at a ratio of 1:6 v/v followed by overnight incubation at 4 °C. Phages in suspension was pelleted by centrifugation at 5000 rcf for 45 min and resuspended in 10 mL of RO Millipore water and subjected to mild agitation at 4 °C for at least 2 h. PEG-purified samples went through further purification using Amicon Ultra centrifugal filters

(Millipore Sigma, Ultra-15, MWCO 10 KDa, and 3 KDa). ATP background signal was measured at different stages of purification and compared with starting phage suspension to monitor the change in background noise. For all liquid phage infection assays, the initial titer of each phage was determined using the double agar layer method by three independent experiments.

### XTT metabolic assay

XTT assay is commonly used to measure metabolic activity of cells using a tetrazolium salt which is a formazan compound in the presence of metabolically active cells followed by a detectable change of color. Assays were conducted in clear, flat-bottom, 96-well plates with total volume of 100 µL in each well. In all, 12 µL of phage suspensions ($-10^9-10^5$ PFU/mL) were mixed with 38 µL bacteria at $OD_{600}$ -0.1 and added to wells at the final multiplicity of infection (MOI) of 10, 1, 0.1, 0.01, and 0.001. XTT solution was prepared in LB media to contain 0.2 mg/mL of XTT and 0.1 mM menadione. 50 µL of the XTT solution was added to all wells, followed by the addition of 50 µL of LB media as negative control, 12 µL of media with 38 µL of bacterial suspension as positive control, and 12 µL of phage with 38 µL bacterial suspension as phage-infected sample. Absorbance was measured at 490 nm wavelength every 5 min for at least16 h using Synergy Neo2 BioTek plate reader set at 37 °C ($n = 3$). The absorbance value from wells containing XTT solution with LB media was subtracted from bacteria-containing wells.

### Optical density assay ($OD_{600}$)

The overnight culture of bacterial strain was added to fresh LB or TSB media at 1:100 v/v ratio and incubated until the sub-culture reached $OD_{600}$ -0.1. Assays were conducted in transparent flat bottom 96-well plate with total assay volume of 200 µL per well. 45 µL of Phage solutions ( $-10^9$ to $10^5$ PFU/mL) with 155 µL of bacteria at $OD_{600}$ -0.1 ($-3 \times 10^7$ CFU/mL) were added to specific wells at the final multiplicity of infection (MOI) of 10, 1, 0.1, 0.01, and 0.001. Control wells contained 155 µL bacteria and 45 µL bacterial media. Absorbance was measured at 600 nm wavelength using Synergy Neo2 BioTek plate reader ($n = 3$). Data was collected every 5 min for at least 16 h at 37 °C. $OD_{600}$ numbers was calculated by subtracting the absorbance of media, divided by the pathlength correction obtained from BioTek plate reader.

### Transmission electron microscopy

Phages with titers of $10^9$ PFU/mL were absorbed onto plasma-cleaned carbon-coated copper grids and negatively stained with 1% uranyl acetate. Stained grids were dried at room temperature and imaged using Talos L120C transmission electron microscope at the Canadian Centre for Electron Microscopy (CCEM), McMaster University (ON, CA).

### ATP Bioluminescence assay

ATP assay reagent solution was by reconstituting the lyophilized ATP reagent solution using bioluminescence Assay Kit CLS II (Sigma-Aldrich, Oakville, ON) based on manufacturer's instruction. For conducting ATP assay in pullulan-trehalose sugar solution, lyophilized ATP reagents of the ATP bioluminescence Assay Kit CLS II were reconstituted with the sterile sugar solution containing 10 wt% pullulan and 0.5 M trehalose, at the same liquid volume as manufacturer's instruction.

Assays were conducted in white flat bottom 96-well plate with total assay volume of 100 µL per well. Bioluminescence signal was detected from wells containing 50 µL of ATP reagent solution mixed with 50 µL of the sample (phage infected or uninfected bacterial suspension). In all, 12 µL of phage suspensions with -$10^9$ to $10^5$ PFU/mL and 38 µL of bacteria at $OD_{600}$ -0.1 ($-3 \times 10^7$ CFU/mL) was used for obtaining MOIs of 10, 1, 0.1, 0.01, and 0.001. The ATP reagent solutions were added at the end point to samples and the signal was

measured at room temperature. The RLU signal was measured at various times points (0, 30, 60, 120, and 180 min) using Synergy Neo2 BioTek plate reader ($n = 3$). The ATP standard curves were prepared per manufacturer's instruction to calculate the ATP amounts.

For one-pot ATP bioluminescent assay, 50 µL of ATP reagent solution was added to specified wells, followed by 12 µL of phage suspensions at different concentrations ($10^9-10^5$ PFU/mL) and 38 µL bacterial sub-cultures at $OD_{600} = 0.1$ respectively, to obtain specific MOIs (10, 1, 0.1, 0.01, and 0.001) within a final volume of 100 µL per assay ($n = 3$).The plates containing ATP assay reagent solution, phage-infected, and uninfected bacterial cultures were incubated at 37 °C inside a Synergy Neo2 BioTek plate reader and the RLU signal was recorded every 5–10 min in stationary state without shaking to prevent damaging the sensitive ATP reagents. Wells with uninfected bacterial suspensions, and wells containing phage only were used as controls. The RLU values from phage-only wells were subtracted from the data when further dilution of the samples would lead to significant titer loss.

The integration time was kept constant at 5 s for all the endpoint and one-pot ATP bioluminescence measurements.

### Stabilization in sugar polymer matrices

Sugar solutions containing 10 wt% pullulan and 0.5 M trehalose were dissolved in milli-Q water and autoclaved to sterilize.

### ATP bioluminescence assay reagent stability

To assess the stability of ATP bioluminescence assay reagent in sugar solution, the ATP assay reagents were prepared in 10 wt% pullulan and 0.5 M trehalose, and the activity was tested with ATP standard solution at 37 °C. Three concentrations of ATP standard solution (0.4, 0.01, 0.001 µM) were prepared in water. To evaluate the ATP reagent solution's activity, both with and without sugar polymers, four conditions were examined: immediate testing after preparation at room temperature (no prior exposure at 37 °C), testing 1 h after incubation at 37 °C, testing 3 h after incubation at 37 °C, and testing 6 h after incubation at 37 °C. Subsequently, ATP standard solutions were introduced into the treated and untreated ATP reaction solution, in the presence and absence of the sugar mixture. The signal was initially measured at time zero, followed by continuous measurements every 10 min for up to 3 h to comprehensively compare the rate of signal decay in the absence and presence of sugars ($n = 3$).

### Phage stability

Phage suspensions were diluted in sugar solution to achieve a titer of $10^9$ PFU/mL, and 100 µL of the phage suspension were added to 24-well plates and air-dried. Stability of phages within dried pullulan-trehalose matrix was tested. Phage titer was calculated 1-, 7-, and 30-days after storage at room temperature using the plaque overlay assay method ($n = 3$).

### All-inclusive dried tablet-based ATP assay

To stabilize the ATP reagents and the phage particles in a dried format, the plate containing phages (15 µL of $10^9$ PFU/mL) and ATP reagent solution (50 µL) in sugars solution was added to specified wells. The plate was dried under nitrogen flow in a glove bag for at least 4 h, followed by storage under vacuum at −0.08 MPa. The Stability of dried tablets containing ATP reagent solution and different phages in 96-well-plate was assessed after one- and 4-weeks storage at room temperature. For conducting the assay, the dried tablet containing the ATP reagents and phage particles embedded in sugar polymer matrix were rehydrated using 50 µL of sterile milli-Q water, followed by addition of 50 µL bacteria sub-culture at $OD_{600} = 0.1$ ($-3 \times 10^7$ CFU/mL), and reading the signal every 5 min at 37 °C. As a control, a plate containing phages and ATP reagent solution was prepared without addition of sugar polymers, and dried/stored under same conditions. The plate

was rehydrated and tested following the same process used for assessing the phage-ATP reagent solution encased in sugar matrices.

### Phage library screening using all-inclusive ATP bioluminescence assay in sugar polymer matrices

To screen the clinical isolates of *P. aeruginosa*, including C0072 and C0335, the in-house phage library including 16 phages from different species (*Pseudomonas* phages including P32, JG004, PP7, E79, PO4, and Phi6; *staphylococcus aureus* phages including phage K and Remus, *E. coli* phages including T7, MS2, HK97, PR772, Lambda, PHIX174, phage 5 and MU, and *Listeria* P100 phage) was used. 10 μL of each phage was mixed with 50 μL of ATP reagent solution containing 10 wt% pullulan and 0.5 M trehalose in wells of a white flat-bottom 96-well plate. The control well for uninfected bacterial control contained 50 μL of sugar-based ATP reagent solution and 10 μL of LB media. The assay reagents along phages were desiccated under nitrogen flow in a glove bag for at least 4 h, followed by storage under vacuum at −0.08 MPa for 1 week. On the day of assay, tablets were rehydrated with 50 μL sterile milli-q water, followed by addition of 40 μL bacteria subcultures of each clinical strain (including C0072 and C0335) grown in LB media at $OD_{600}$ = 0.1 (~$3 \times 10^7$ CFU/mL), and reading the signal every 5 min for up to 6 h at 37 °C. All the phages had an approximate titer in the range of $10^8$–$10^9$ PFU/ml. (*n* = 3).

To screen the library of *Salmonella* phages, 10 μL of each phage from the library (Supplementary Table 2) with initial titers above ~$10^8$ PFU/mL and 10 μL of TSB media for negative control wells (uninfected bacterial culture) were added to 50 μL of ATP reagent solution containing 10 wt% pullulan and 0.5 M trehalose in wells of a white flat-bottom 96-well plate. Afterwards, 40 μL of *Salmonella* bacterial subculture of (S. Newport C487 and S. Senftenberg S-219/89) grown in TSB media to $OD_{600}$ = 0.1 were added to each well. The plates were incubated at 37 °C inside a Synergy Neo2 BioTek plate reader and the RLU signal was recorded every 6 min in stationary state without shaking to prevent damaging the sensitive ATP reagents. (*n* = 3).

To screen the *E. coli* phage library against LF82 and O157:H7, we followed the same screening steps as those employed for the salmonella phage library. 40 μL of sub-cultures of *E. coli* LF82 and O157:H7 grown in LB media at $OD_{600}$ = 0.1 were added to wells containing 50 μL of ATP reagent solution containing 10 wt% pullulan and 0.5 M trehalose, and 10 μL of each phage in wells of a white flat-bottom 96-well plate. The list of *E. coli* phages can be found in Supplementary Table 3. The RLU signals were read every 10 min at 37 °C using a Synergy Neo2 BioTek plate reader. (*n* = 3).

Similar to other strains, to screen the S. aureus phage library, 40 μL of sub-cultures of *S. aureus* 68 HER 1049 and *S. aureus* 44 A HER 1101 strains grown in TSB media at $OD_{600}$ = 0.1 were added to wells containing 50 μL of ATP reagent solution containing 10 wt% pullulan and 0.5 M trehalose, and 10 μL of each phage in wells of a white flat-bottom 96-well plate. The list of *S. aureus* phages can be found in Supplementary Table 4. The RLU signals were read every 10 min at 37 °C using a Synergy Neo2 BioTek plate reader. (*n* = 3).

### Modeling conditions for lysis from without

Three *P. aeruginosa* strains namely PAO1 (Pa), C0072 and C0335 (Supplementary Table 1) were infected at MOIs 10000, 1000, 100, 10, 1, 0.1 and 0.01. For spot test, JG004 phage with initial concentration of ~$5 \times 10^{11}$ PFU/mL was serial diluted (10-fold) from a dilution factor of 1 to $10^{-8}$ and 10 μL of each dilution was spotted on a bacterial lawn of *P. aeruginosa* strains. For ATP bioluminescence assay, 40 μL bacterial sub-cultures at $OD_{600}$ = 0.1 (~$10^7$ CFU/mL) was added to wells of a 96-well plate containing 50 μL of ATP reagent solution in 10 wt% pullulan and 0.5 M trehalose, mixed with 10 μL of phage suspensions at different concentrations ($5 \times 10^{11}$–$5 \times 10^5$ PFU/mL), to obtain specific MOIs (10000, 1000, 100, 10, 1, 0.1 and 0.01) within a final volume of 100 μL per assay (*n* = 3). The plate was incubated at 37 °C inside a Synergy

Neo2 BioTek plate reader and the RLU signal was recorded every 5 mins. Optical density ($OD_{600}$) assay was conducted at the same MOIs by adding 80 μL of bacterial subcultures at $OD_{600}$ = 0.1 (~$10^7$ CFU/mL) to 20 μL of phage at different concentrations ($5 \times 10^{11}$–$5 \times 10^5$ PFU/mL) and 100 μL of LB media to a final volume of 200 μL to obtain MOIs of 10000, 1000, 100, 10, 1, 0.1 and 0.01 while keeping the final phage/bacteria concentration same as the one-pot ATP bioluminescence assay (*n* = 3). The absorbance at 600 nm was recorded at 37 °C every 5 min using inside a Synergy Neo2 BioTek plate reader. Wells with uninfected bacterial suspensions were used as controls in both assays.

### Reporting summary

Further information on research design is available in the Nature Portfolio Reporting Summary linked to this article.

## Data availability

The data generated in this study are provided in Source Data file. Source data are provided with this paper.

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

## Acknowledgements

This research was undertaken, in part, thanks to funding from the Canada Research Chairs Program (S.M. (grant no. 950-232136), T.F.D.(950 – 233115), and Z.H.(CRC-2020-00096)). T.F.D. (grant no. RGPIN-2024-06761), Z.H. (grant no. RGPIN-2016-05605), and C. D. M. F. (RGPIN-06354-2018) acknowledge funding from Natural Sciences and Engineering Research Council of Canada (NSERC) Discovery Grants Program. Z.H. (grant no. ER22-17-067) and T.F.D. (grant no. ER18-14-233) acknowledge funds from Ontario Early Researcher Awards. The transmission electron microscopy was carried out at the Canadian Center for Electron Microscopy (CCEM), a national facility supported by the NSERC and McMaster University.

## Author contributions

F.B. conceived the study, designed, and executed the experiments, analyzed the data, prepared the figures, and wrote the initial draft of the manuscript. A.H. made important contributions to phage stability testing, XTT and optical density assays. M.T. optimized the XTT assay protocol, contributed to discussion, and revision of the manuscript. C.D.M.F. provided scientific editing based on expertize on biomolecule preservation. D.T. prepared the phage libraries as well as hosts for all the *Salmonella*, and *S. aureus* phages. S.M. provided phages, scientific editing, and feedback on data analysis. Z.H. and T.F.D. conceptualized and supervised the project, and guided the experimental design, data analysis and manuscript writing.

## Competing interests

The patent for using pullulan/trehalose to stabilize biological agents has been licensed to Elarex. Carlos Filipe is a co-founder and currently a scientific advisor to Elarex. A patent application has been filed by McMaster University (inventors F.B, T.F.D, Z.H.) to the Canadian Intellectual Property patent office pertaining to the high throughput aspect(s) of this work (application number PCT/CA2024/050748). The remaining authors declare no competing interests.
