## [Peer Review File · Nature Communications]

REVIEWER COMMENTS

Reviewer #1 (Remarks to the Author):

GENERAL COMMENTS

This review is based on more than 50 years of developing assays based on firefly luciferase, especially continuous monitoring of ATP. I have, however, limited experience with bacteriophages.

Key results: Antibiotic resistance is a rapidly growing problem in healthcare today. The use of bacteriophages to treat bacterial infections is a potential way to solve the situation. A prerequisite is that there are simple and rapid techniques for rapid target identification of suitable bacteriophages in personalized phage therapy. The present manuscript is a valuable contribution for this purpose.

Validity: The one-pot bioluminescence assay fulfills the prerequisite stated above as it only requires the following items:

- A. A library of phages dried in an ATP bioluminescence reagent with stabilizers of both the reagent and the phages.
- B. A sample containing the bacterial strain in a mid-exponential phase.
- C. A microplate luminometer temperature controlled at 37 °C.

Monitoring bacterial lysis with an ATP bioluminescence reagent is ideally performed under the following conditions:

1. The ATP reagent should have a low luciferase level since the luciferase reaction degrades ATP.
2. ATP degrading enzymes from the lysed bacterial cells should be inhibited or so strongly diluted in the lysis that they only have a non-significant effect on the signal.
3. Luciferase or luciferin should not be gradually inactivated during the measurement.

Under conditions 1-3 the light emission will be a direct measure of the lysis. Comparing Figure 3A (ii) with 4D and 4E indicates that this has been at least partially achieved in the one-pot bioluminescent assay. Calibrating the assay by addition of a known amount of ATP standard a) at the start of the measurement, b) before the peak, c) at the peak and d) at the end of the measurement will tell how close the light signal follows the lysis. The additions should be made in four different wells and in a small volume compared to the reaction mixture but at least a 10x higher amount of ATP compared to the reaction mixture. Please cf. section 9 in attached paper.

Significance: The one-pot bioluminescent ATP assay represents a major improvement in the field.

Data & methodology: For the end-point ATP bioluminescent assays, ATP concentrations were calculated in nM by referring to ATP standard curves according to manufacturer's instructions. This is acceptable provided one is sure that the luciferase activity is the same in all samples. Luciferase

can be inhibited by inhibitory compounds in the samples and by aging. In the one-pot ATP bioluminescent assay it would not have been correct to refer to ATP standard curves as one can't be sure that the luciferase activity is the same in all samples.

The tablets containing ATP reagent, bacteriophages and stabilizers can be used after one week at room temperature. Is there any stability data obtained at lower temperatures? Long-term stability will be required when building up libraries with different bacteriophages. Lyophilized ATP reagents from some manufacturers are stable for 2 years or more at +4 °C and -18 °C.

The manuscript does not state which instrument was used for the bioluminescent measurements. I suppose it was the Synergy Neo2 BioTek plate reader. Is this correct? Were microplates shaken in any of the assays?

Conclusions: The stability of the tablets is promising, but more detailed studies at different storage temperatures and longer durations will be needed as the tablets will be a medical device.

Suggested improvements: The detailed stability studies is most likely performed as part of the commercialization of the tablets and must not be included in the present manuscript.

References: References are appropriate.

Clarity and context: This is an excellent paper requiring only minor modifications.

SPECIFIC COMMENTS

Lines 1, 8 and 57: The words "Portable" and "Ultraportable" are true with respect to the reagents, but a temperature-controlled microplate reader is not an easily portable instrument (Synergy Neo2 BioTek plate reader weighs 35 kg). Please find another word or define what you mean by portable.

Lines 61-68: Please specify the advantages of ATP compared to leakage of other intracellular components or to monitoring absorbance as in Figure 1D?

Line 89: Is "the burst of ATP" similar to what would be expected from leakage of 100 % of the normal intracellular ATP concentration in the same number of uninfected cells?

Lines 117-118: It is assumed that bioluminescence should be deleted in "XTT bioluminescence assay".

Lines 155-158: There are techniques for reducing the ATP background to less than 0.0000025 nmol/L.

Line 159: Point rather than "pint".

Lines 262-264: What is the negative control in the sentence: "the signal at 6 hrs post infection in cultures infected with P32 (Figure 4B) and JG004 (Figure 4C) containing pullulan-trehalose mixture was ~ 99% and 95% higher, respectively, than the negative control".

Lines 267-269: "As shown in Figure S4, titer loss for phage encased in the sugar polymer matrix was less than one log for all three phages after a 30-day storage under ambient condition." A 30-day

stability in ambient conditions indicates that the stability at +4 °C or -18 °C may be good enough. How about the stability of the ATP reagent?

Line 276: Should “Figure 4B” be Figure 4D?

Lines 279-283: “Lastly, it should be noted that drying the ATP reagents and phage in the absence of sugar polymers led to complete loss of signal after the one-week storage period, as shown in Figure 4F. This is particularly advantageous for ease of transportation and point-of-use accessibility of our phage screening technology, especially in remote regions.” The two sentences do not match.

Line 294: Should “B, E and F” be D, E and G?

Line 296: Should “C and D” be B and C?

Lines 435-436: “To normalize the results, absorbance values from wells containing XTT solution with 436 LB media was subtracted from bacteria-containing wells.” Normalize normally means dividing. It is suggested to delete “To normalize the results”.

Dr. Arne Lundin

arne.lundin@biothema.com

Reviewer #3 (Remarks to the Author):

The authors report a platform technology that is potentially portable and high-throughput for the rapid identification of sensitive phages for clinical bacterial pathogens and thus may be helpful in personalized phage therapy. This technology was created on the basis of detecting the bioluminescence signal generated by luciferin, luciferase, and ATP in order to identify ATP molecules released during phage-mediated bacterial lysis. Additionally, they prepared a mixture of phage suspension and lyophilized ATP detection reagents for longer storage using pullulan and trehalose. Finally, they used this technology to screen phages for clinical *P. aeruginosa* isolates. Overall, the manuscript is generally well written, and easy to follow. However, this reviewer has read the paper carefully and has concerns about the novelty of this technology and the significance of this technology for application potential in the clinic. Luminescent ATP detection has been widely used to examine active cells. Only *Pa* isolates were tested in this study. Please see the major comments below:

1) This study was designed to reduce background ATP signal in phage suspensions by roughly diluting the suspension. Only 3 phages were tested on a *Pa* strain. Different phages have different

features in infecting even the same host. It is not clear to this reviewer if nonlytic bacterial cells can release ATP. The background may lead to wrong conclusions.

2) The optimal MOI and detection time point vary for different bacterial strains and phages and thus can impact the final signal detection results, suggesting that this technology may not be fit for large-scale screening of phages for various bacterial strains.

3) In addition to Pa isolates, more clinical bacterial strains (e.g. *Klebsiella pneumoniae*, *Acinetobacter baumannii*, et al.) are needed to test on this technology platform.

4) To my understanding, compared to this technology, the sensitive phages can be screened by simply detecting killing curves within several hours. It is not clear the efficiency of this method only if authors can compare them.

Minor comments:

1. Lines 104-206, Podoviridae and Myoviridae have been discarded by the ICTV. Please update.
2. Line 227. This hypothesis in vitro may not be correct in the real-life application of phage therapy in vivo.
3. Line 230. This claim may be not correct.
4. Figure 4D and Figure 4G were not cited. Line 276, Figure 4B was wrongly cited.
5. Line 318, How to define the efficiency of plaquing the identified phage?

List of Changes Made to the Revised Manuscript:

1. The portable was removed from title (Reviewer 1, Minor Comment #1).
2. Additional data was provided in new ***Supplementary Fig. 4*** and ***Supplementary Fig. 5*** to support our claim about ATP signal representing phage-mediated bacteria lysis. Experimental details and discussion were added to **Page 31, Line 563-573** and **Page 15, Lines 276-300**, respectively (Reviewer 1, Comment #1).
3. Additional data was provided in new ***Supplementary Figure 7*** to support our claim about tablet stability and discussion was added to **Page 17, Line 309-317** (Reviewer 1, Comment #5 and Reviewer 1, Minor Comment #8).
4. Additional data was provided in new ***Figure 5 d-g, Supplementary Fig. 11, and Supplementary Fig. 12*** presenting utility of our method with four more bacterial strains and a significantly larger phage library provided by the Felix D'Herelle Reference Center for Bacterial Viruses. All new bacteria are presented in new ***Supplementary Table 1***. All new phages are presented in new ***Supplementary Table 2***, and ***Supplementary Table 3***. Experimental detail was added to **Page 32, Line 591-618** and discussion was added to **Page 21, Lines 374-388** (Reviewer 2, Comment #1, #3, #4 and Reviewer 1, Minor Comment #2).
5. Additional information was provided in new ***Supplementary Table 4*** and ***Supplementary Table 5*** to benchmark our method against commonly used culture methods (Reviewer 2, Comment #2).
6. Nomenclature was updated based on latest ICTV guidelines (*Revised manuscript, page 7, line 111-112*) (Reviewer 2, Minor Comment #1).
7. A revised description was provided on measures of phage lytic activity (*Revised manuscript, Page 13, Line 230-233*) (Reviewer 2, Minor Comment #3).
8. Method section was amended with details on plate reader use (*Revised manuscript, page 28, line 544-558*) (Reviewer 1, Comment #4).
9. Typographic errors have been corrected (*Revised manuscript, page 7, line 125; page 9, line 166; page 17, line 315; page 17, line 321; page 19, lines 330-332; page 15, line 270; Figure 1d, the y axis title; page 15, line 268; page 17, line 320*) (Reviewer 1, Minor Comment #4, #6, #9, #10, #11, #12, #13; Reviewer 2, Minor Comment #4).
10. Format correction (responding to editorial guidelines):
 - The format of all figures was revised to fit the requirement of Nature Communications.
 - All figure captions were updated based on nature communication formatting instructions in the revised manuscript and SI
 - All figure legends were updated with information about exact sample size, box plot and error bar.
11. Source data file have been provided responding to editorial guidelines.

Reviewer 1

This review is based on more than 50 years of developing assays based on firefly luciferase, especially continuous monitoring of ATP. I have, however, limited experience with bacteriophages.

Key results: Antibiotic resistance is a rapidly growing problem in healthcare today. The use of bacteriophages to treat bacterial infections is a potential way to solve the situation. A prerequisite is that there are simple and rapid techniques for rapid target identification of suitable bacteriophages in personalized phage therapy. The present manuscript is a valuable contribution for this purpose.

Our response:

We sincerely thank the reviewer for taking time to assess our manuscript and provide valuable comments, which was helpful in enhancing the quality of our work.

Comment #1: Validity: The one-pot bioluminescence assay fulfills the prerequisite stated above as it only requires the following items:

- A. A library of phages dried in an ATP bioluminescence reagent with stabilizers of both the reagent and the phages.
- B. A sample containing the bacterial strain in a mid-exponential phase.
- C. A microplate luminometer temperature controlled at 37 °C.

Monitoring bacterial lysis with an ATP bioluminescence reagent is ideally performed under the following conditions:

1. The ATP reagent should have a low luciferase level since the luciferase reaction degrades ATP.
2. ATP degrading enzymes from the lysed bacterial cells should be inhibited or so strongly diluted in the lysis that they only have a non-significant effect on the signal.
3. Luciferase or luciferin should not be gradually inactivated during the measurement.

Under conditions 1-3 the light emission will be a direct measure of the lysis. Comparing Figure 3A (ii) with 4D and 4E indicates that this has been at least partially achieved in the one-pot bioluminescent assay.

Calibrating the assay by addition of a known amount of ATP standard a) at the start of the measurement, b) before the peak, c) at the peak and d) at the end of the measurement will tell how close the light signal follows the lysis. The additions should be made in four different wells and in

a small volume compared to the reaction mixture but at least a 10x higher amount of ATP compared to the reaction mixture. Please cf. section 9 in attached paper.

Our response:

The reviewer highlights a very important point because one of the central hypotheses of our work is that the light emission signal is a direct measure of bacterial lysis. Based on the reviewer's suggestion, we have performed additional experiments and added new data (new **Supplementary Fig. 4** and **Supplementary Fig. 5**) to the **revised manuscript** (and reproduced below) that support our previous data. The experimental design has been added to Methods section (**Revised manuscript, page 31, lines 563-573**) and a description of the results and brief discussion of data has been added to **revised manuscript, page 15, lines 276-300**.

By comparing **revised manuscript Figures 4d** and **Figure 3b**, we clearly show that the addition of a sugar mixture to the ATP reagent solution prevented the deactivation of the luciferase enzyme at 37°C. The **revised Figures 4b and 4d** also revealed that the peak signal in the ATP reaction solution, corresponding to one hour after the start of the assay, was similar in the presence or absence of the sugars. However, the signal sharply dropped after an hour in the absence of sugars, while the presence of sugars prevented such drop.

To show how close the RLU signal follows the phage-mediated cell-lysis, we designed an experiment with ATP standard solutions as suggested. ATP reaction solutions were prepared in the absence and presence of the sugar polymer mixture. ATP standard solutions with three different concentrations (0.4, 0.01, and 0.001 μM) were prepared including in water. The 0.4 μM ATP standard solution was chosen due to RLU values at least 10 times larger than the amount of ATP released in the solution for the phage-mediated ATP detection in our one-pot assays as suggested. The other two additional concentrations of 0.01 and 0.001 μM ATP standard solutions were also tested to see if at lower ATP values we are still able to detect consistent signal in real-time phage mediated RLU signal monitoring at 37°C for weak lysis activity. The activity of the ATP reagent solution with and without sugar polymers were tested in three conditions, namely:

- No heat exposure: used right after preparation at room temperature
- One hour heat exposure: used after one hour of incubation at 37°C (approximately where we detect a peak signal in **revised manuscript, Figure 3b**)
- Three hours heat exposure: used after three hours of incubation at 37°C (after the peak)

ATP standard solutions were added to the above-mentioned three conditions of ATP reaction solutions and the signal was measured at time zero, followed by continuous measurements for up

to 6 hours. The results (**revised manuscript Supplementary Fig.4**) confirmed that sugar polymer solutions were able to stabilize the ATP bioluminescence assay enzymes at 37°C, and the loss of signal was not significant after heat exposure at 37°C. On the other hand, the ATP reagent solution in the absence of sugars led to significant decrease in RLU signal after incubation at 37°C, especially after 3 hours (**revised manuscript supplementary Fig. 5**). In addition, the kinetics of the RLU signal confirmed that the rate of signal loss was also higher when no sugar was present in the ATP reagent solution. Specifically, the loss of signal after 3 hours of exposure to heat at 37°C resulted in approximately a 10-fold reduction in the signal times higher signal loss. This shows that the ATP bioluminescence assay in the presence of sugar polymers is able to monitor the real-time ATP release following phage-mediated bacterial cell lysis.

The provided reference has been cited in the revised manuscript and the following text has been added to the revised manuscript (**Revised manuscript, page 15, lines 276-300**):

“To further assess how closely the RLU signal follows the phage-mediated cell-lysis, we designed an experiment with ATP standard solutions. ATP reaction solutions were prepared in the absence or presence of the sugar polymer mixture. ATP standard solutions with three concentrations (0.4, 0.01, 0.001 μM) were prepared in water. The RLU values correlated with 0.4 μM ATP standard solution was chosen at least 10 times larger than the amount of ATP released in the solution for the phage-mediated ATP detection in our one-pot assays. 0.01 and 0.001 μM ATP standard solutions were tested to see if at lower ATP values we are still able to detect consistent values in real-time phage mediated RLU signal monitoring at 37°C for weak lysis activity. The activity of the ATP reagent solution with and without sugar polymers were also tested in three conditions including no previous exposure at 37°C (tested right after preparation at room temperature), one hour after incubation at 37°C, three hours after incubation at 37°C. ATP standard solutions were added to the above-mentioned treated and untreated ATP reactions in the presence and absence of the sugar mixture, and the signal was measured at time zero, followed by continuous measurements for up to 6 hours. The results confirmed that sugar polymer was able to stabilize the ATP bioluminescence assay at 37°C, and the loss of signal was not significant after exposure at 37°C in all three ATP standard concentrations. On the other hand, in the absence of sugars, the ATP reagent solution led to a significant decrease in RLU signal after incubation at 37°C, especially after 3 hours (**Supplementary Fig. 4**). In addition, the kinetic RLU signal monitoring showed that the rate of signal loss was also higher when no sugar was present in the ATP reagent solution (**Supplementary Fig. 4**). Specifically, the loss of signal after 3 hours incubation at 37°C resulted in an approximately 10-fold signal loss (**Supplementary Fig. 5**). This confirms that ATP bioluminescence assay in the presence of sugar polymers has been able to detect the ATP release as a result phage-mediated bacterial cell lysis.”

Supplementary Figure 4. Effect of sugar polymers on stability of the ATP reagent solution at 37°C. ATP standard solutions were added to ATP reagent solution with (+) and without (-) sugar mixture at time =0, after incubating for one and three hours at 37°C, and the results were compared to no heat exposure ATP reagent solution. ATP standard solutions are used at **a** 0.4 μM , **b** 0.01 μM , and **c** 0.001 μM . The RLU signal was measured kinetically every 10 minutes for up to 6 hours. Sugar mixture was able to preserve the activity of luciferin and luciferase as there were no significant changes in the RLU signals after heat exposures. In the absence of sugars, the activity of ATP reagent solution decreased as significantly lower RLU signals were detected, which was more drastic after 3 hours of heat exposure. The RLU signal also dropped faster in the absence of

sugar mixture. The presented data are the average of three replicates (n=3) with error bars representing standard deviation from the mean.

Supplementary Figure 5. Comparison of the RLU signal intensity of heat treated and untreated ATP reaction solutions in the presence and absence of sugar mixture after heat exposure at 37°C. RLU signal at time=0, right after adding ATP standard solution with 0.4 μM, 0.01 μM, 0.001 μM to ATP reagent solutions with (+) and without (-) sugar mixture after incubating for 1 and 3 hours at 37°C in comparison with no heat exposure ATP reagent solution (time=0). The presented data are the average of three replicates (n=3) with Standard deviation from the mean. Statistical significance in all panels is derived from Two-way analysis of variance (ANOVA). Significance levels include * $P < 0.05$, ** $P < 0.01$, *** $P < 0.001$, and **** $P < 0.0001$.

Comment #2: Significance: The one-pot bioluminescent ATP assay represents a major improvement in the field.

Data & methodology: For the end-point ATP bioluminescent assays, ATP concentrations were calculated in nM by referring to ATP standard curves according to manufacturer's instructions. This is acceptable provided one is sure that the luciferase activity is the same in all samples. Luciferase can be inhibited by inhibitory compounds in the samples and by aging. In the one-pot ATP bioluminescent assay it would not have been correct to refer to ATP standard curves as one can't be sure that the luciferase activity is the same in all samples.

Our response:

The reviewer highlighted a key point. This is why we do not report the exact amount of ATP released, but rather the RLU signal. However, our data clearly shows that the one pot assay is working well in detecting the phage-mediated cell lysis and that the RLU signal in continuously measured for up to 6 hours in the presence of sugar mixture (**Revised manuscript, Figure 4d**). We also confirmed that the presence of sugars (pullulan and trehalose) has a critical effect on the thermal stability of the assay reagents with our new data (**Revised manuscript, Supplementary**

Fig. 4, and Supplementary Fig. 5). Furthermore, the shape of the 6 hr RLU kinetic curves are different for each phage-bacteria pairs, which is expected considering the different phage latent periods. Specifically, the signal starts to increase at different time points in one-pot ATP assay in the presence of sugar polymers which confirms active enzymes and phage-mediated cell lysis happening in the solution.

Comment #3: The tablets containing ATP reagent, bacteriophages and stabilizers can be used after one week at room temperature. Is there any stability data obtained at lower temperatures? Long-term stability will be required when building up libraries with different bacteriophages. Lyophilized ATP reagents from some manufacturers are stable for 2 years or more at +4 °C and -18 °C.

Our response:

Lyophilized ATP reagents can be stored for an extended period of time, but to use them for phage screening, we would still need to prepare the solutions, add phages, and run the assay, making the process lengthy and laborious. Using desiccated sugar-based tablets containing phage and ATP reagents is more straightforward. The tablets can be prepared in large batches and stored. We have not tested the stability of the ATP reaction solution at lower temperatures, such as 4°C, because one of the objectives of our work was to cut out the cold chain and thus footprint and cost needed for storage of the biobanks and assays, which can be very impactful in underdeveloped countries, remote regions, or disaster scenarios. Although we have not tested the 4°C stability, we do have stability data for storage at room temperature under vacuum up to 4 weeks (**Revised manuscript, supplementary Fig. 6 and 7**) that clearly show the preservation of phage and all-inclusive tablets.

Comment #4: The manuscript does not state which instrument was used for the bioluminescent measurements. I suppose it was the Synergy Neo2 BioTek plate reader. Is this correct? Were microplates shaken in any of the assays?

Our response:

Indeed, we used the Synergy Neo2 BioTek. The measurements were conducted without any shaking as we have reason to believe that continuous shaking will damage the sensitive ATP

assay components. In fact, the manufacturer's instructions strongly advise against vortexing and call for slow manual swirling of the container or gentle aspiration with a micropipette to mix the reagents. This information has been added to the manuscript and the methods section was updated to read:

Revised manuscript, Page 28, Line 544-547:

“For end-point ATP bioluminescent assay, the ATP reagent solutions were added at the end point to samples and the signal was measured at room temperature. The RLU signal was measured at various times points (0, 30, 60, 120, and 180minutes) using the Synergy Neo2 BioTek plate reader.”

Revised manuscript, Page 28, line 552-558:

“The plates containing ATP assay reagent solution, phage-infected, and uninfected bacterial cultures were incubated at 37°C inside a Synergy Neo2 BioTek plate reader and the RLU signal was recorded every 5-10 minutes in stationary state without shaking to prevent damaging the sensitive ATP reagents. Wells with uninfected bacterial suspensions, and wells containing phage only were used as controls. The RLU values from phage-only wells were subtracted from the data when further dilution of the samples would lead to significant titer loss.”

Comment #5: Conclusions: The stability of the tablets is promising, but more detailed studies at different storage temperatures and longer durations will be needed as the tablets will be a medical device.

Suggested improvements: The detailed stability studies is most likely performed as part of the commercialization of the tablets and must not be included in the present manuscript.

Our response:

We agree with the reviewer that the detailed stability studies must be performed as part of the commercialization of the tablets. Yet, we e have added new data showing stability for desiccated tablets after 4 weeks of storage at room temperature under vacuum to the revised manuscript (**Revised manuscript, supplementary Fig. S7**, reproduced below). These data will serve as the foundation on which the long-term stability tests will be performed at the pre-commercialization stage with other phages. To reflect the new data, the following text has been added to the **revised manuscript, Page 17, Lines 309-317:**

“Although the signal intensity was lower compared to the original liquid assay, the RLU signal of the phage containing tablets compared to control uninfected Pa was visibly higher showing a

successful detection of the phage-mediated bacterial cell lysis ... The all-inclusive tablets also preserved their stability after four weeks of storage at room temperature under vacuum (Supplementary Fig. 7).”

Supplementary Figure 7. Phage and ATP reagents stability assay after 4 weeks. The all-inclusive tablets including phages and ATP reagent solutions in sugar polymer matrices were stored for 4 weeks at room temperature under vacuum. To run the assay, the tablets were reconstituted with sterile Milli-Q water, followed by addition of *P. aeruginosa* (Pa) subcultures at OD₆₀₀ ~0.1 and kinetic monitoring of RLU signal. The data presented are the average of 6 replicates, and dashed line showing standard deviation from the mean.

Comment #6: References: References are appropriate.

Clarity and context: This is an excellent paper requiring only minor modifications.

Our response:

Thank you, again, for taking the time to provide feedback on our manuscript.

Minor Comments

Comment #1: Lines 1, 8 and 57: The words “Portable” and “Ultraportable” are true with respect to the reagents, but a temperature-controlled microplate reader is not an easily portable instrument

(Synergy Neo2 BioTek plate reader weighs 35 kg). Please find another word or define what you mean by portable.

Our response:

This is a valid point. We have removed portable from the title and have moved the word “portable” to **revised manuscript, page 1, line 8**, where we are explaining the characteristics of phage-ATP assay reagent tablets. We have also revised the sentence on **revised manuscript, page 3, line 64** so that the word portable refers to assay reagents:

“We envision a platform technology that addresses both aforementioned challenges in personalized phage therapy, namely a reliable high throughput technology that can be implemented rapidly with shelf-stable and portable assay reagents that can be readily shipped around the world to be employed at the point of care with minimal infrastructure or training.”

We would like to highlight the fact that the word “portable” refers to the proposed format for phage biobanks, which is in the format of microtiter plates containing stable solid tablets of phages plus the detection biochemistry. It can be stored on the shelf and shipped around the world in solid format and with no need for a cold chain. We envision our method to stabilize phage biobanks such that multiple copies can be made and stored to be shipped to any international hospital/research lab with very short notice and in a safe and cost-effective manner. This contrasts with the current status, where biobanks often supply phages in the format of vials of liquid phage lysate stored in the fridge, thus taking up a much larger footprint. In addition, these phages may need to be amplified before shipping, time that is costly for a patient with a resistant infection.

Comment #2: Lines 61-68: Please specify the advantages of ATP compared to leakage of other intracellular components or to monitoring absorbance as in Figure 1D?

Our response:

A. Comparing ATP to leakage of other intracellular components: While it is true that in theory, other intracellular components could be used for detection of bacterial cell lysis, ATP does offer advantages for real time application. ATP is present in all living organisms and measuring this biomolecule is straightforward. There are also well-established commercial ATP kits available that can be easily used.

B. Comparing ATP to absorbance (OD measurements): In the revised manuscripts, we have provided kill curves for all the phage-bacteria pairs. Please refer to **revised manuscript, Supplementary Fig. 8, 11, and 12**, reproduced below. We have also added this description to **revised manuscript, page 22, lines 396-404:**

“What gives the ATP detection method an edge over other components is that it relieves a *major bias* shared by optical density and metabolic activity monitoring, namely detection of slowed growth not accompanied by bacterial lysis (which happens for chronic phage or in some cases of lysogeny) vs bacterial cell lysis and destruction. The burst of ATP detected in our method can only happen if the bacterial cell is compromised and thus the detection of phage-mediated ATP release *directly* detects cell lysis, while other methods measure infectivity *indirectly* by monitoring the phenotypic features of bacterial cultures. Moreover, ATP release shows a real time measurement of bacterial cell lysis which can provide information on phage dormant period on a particular bacterial strain.”

Supplementary Figure 8. Screening Clinical *P. aeruginosa* isolates against an in-house phage library. **a** C0072 strain isolated from patient with urinary tract infection screened against a library of phages including phages from different species (**Supplementary Table 1**) using desiccated Sugar-based one-pot ATP bioluminescence assay, optical density assay (OD₆₀₀), and representative spot test on C0072 bacterial lawn. **b** C0335 strain isolated from patient with arm infection, screened against an in-house phage library using one-pot ATP bioluminescence assay optical density assay (OD₆₀₀) and spot test on C0335 bacterial lawn. The OD₆₀₀ assay and spot test was conducted on three *P. aeruginosa* phages including P32, JG004, and PP7. ATP and OD assay were conducted with at least three replicates (n=3).

Supplementary Figure 11. ATP and OD₆₀₀ kinetic curves and spot test of two *Salmonella* strains against 16 *Salmonella* phages. a *Salmonella enterica* serovar Newport. b *Salmonella enterica* serovar Senftenberg. ATP one-pot assay was conducted in the presence of sugar polymers. The ATP and OD₆₀₀ were conducted in triplicates (n=3), spot tests were repeated independently twice. The error bars show standard deviation from the mean.

Supplementary Figure 12. ATP and OD₆₀₀ kinetic curves and spot test of two *E. coli* strains against 28 *E. coli* phages. a *E. coli* O157:H7 strain (human fecal isolate). b LF82 (Crohn's disease isolate¹). ATP one-pot assay was conducted in the presence of sugar polymers. The ATP and OD₆₀₀ were conducted in triplicates (n=3).

Comment #3: Line 89: Is “the burst of ATP” similar to what would be expected from leakage of 100 % of the normal intracellular ATP concentration in the same number of uninfected cells?

Our response:

In theory, yes, if a single bacterial cell is lysed by phages (or if the cell is compromised by any other method), it is expected that all the ATP and other intracellular components will be released into the medium. However, for a bacterial culture, the picture may be more complicated. The theory on phage-bacteria infection dynamic predicts that depending on the conditions (e.g., MOI) only a fraction of bacterial cells will be lysed by phages at any given time. When examining the ATP release kinetic curves (e.g., **Revised manuscript, Figure 4d**) you will see that the ATP released to the solution increases with time and eventually reaches a plateau. Uninfected bacterial cells continue to grow and not release ATP while the phage-infected cells will eventually do.

Comment #4: Lines 117-118: It is assumed that bioluminescence should be deleted in “XTT bioluminescence assay”.

Our response:

Thank you. This was a typo and we have now deleted the word “bioluminescence” from (**Revised manuscript page 7, line 125**) since XTT is a colorimetric assay.

Comment #5: Lines 155-158: There are techniques for reducing the ATP background to less than 0.0000025 nmol/L.

Our response:

Thank you for pointing this out. One of the first experiments we performed was purification of phage stocks through a range of techniques, some simple and some very rigorous to compare the signal to noise ratio. We thought this would be important as biobanks purify phages to different extents and no unified method exists. We found the dilution method to result in a signal to noise ratio comparable to some of the most rigorous purification methods (**Revised manuscript, Supplementary Fig. 3**).

Comment #6: Line 159: Point rather than “pint”.

Our response:

Thank you, typo has been corrected in **revised manuscript page 9, line 166**.

Comment #7: Lines 262-264: What is the negative control in the sentence: “the signal at 6 hrs post infection in cultures infected with P32 (Figure 4B) and JG004 (Figure 4C) containing pullulan-trehalose mixture was ~ 99% and 95% higher, respectively, than the negative control”.

Our response:

The negative control is the uninfected *Pseudomonas* culture which has a very low signal at this timepoint.

Comment #8: Lines 267-269: “As shown in Figure S4, titer loss for phage encased in the sugar polymer matrix was less than one log for all three phages after a 30-day storage under ambient condition.” A 30-day stability in ambient conditions indicates that the stability at +4 °C or -18 °C may be good enough. How about the stability of the ATP reagent?

Our response:

Phages have usually high stability at +4 °C. The stability of the ATP reagent solutions depends on the source. It can be as high as two years at -18 °C in lyophilized state. For ease of use, when mixed with phage, the stability of the liquid mixture of phage and ATP assay reagents was tested at RT (**Revised manuscript, Figure 4e and new Supplementary Fig. 7**).

Supplementary Figure 7. Phage and ATP reagents stability assay after 4 weeks. The all-inclusive tablets including phages and ATP reagent solutions in sugar polymer matrices were stored for 4 weeks at room temperature under vacuum. To run the assay, the tablets were reconstituted with sterile Milli-Q water, followed by addition of *P. aeruginosa* (Pa) subcultures at OD₆₀₀ ~0.1 and kinetic monitoring of RLU signal. The data presented are the average of 6 replicates, and dashed line showing standard deviation from the mean.

Comment #9: Line 276: Should “Figure 4B” be Figure 4D?

Our response:

Thank you. We have corrected the citations of Figure 4 in this section thoroughly and the changes have been highlighted in the text, **revised manuscript page 17, line 315.**

Comment #10: Lines 279-283: “Lastly, it should be noted that drying the ATP reagents and phage in the absence of sugar polymers led to complete loss of signal after the one-week storage period, as shown in Figure 4F. This is particularly advantageous for ease of transportation and point-of-use accessibility of our phage screening technology, especially in remote regions.” The two sentences do not match.

Our response:

Thank you. We have revised the sentences on **revised manuscript page 17, line 320-322** to read:

“This clearly shows the importance of sugar polymers in preserving phages and ATP assay components, which is particularly advantageous for ease of transportation and point-of-use accessibility, including disaster scenarios as well as underdeveloped and remote regions.”

Comment #11: Line 294: Should “B, E and F” be D, E and G?

Our response:

Thank you. We have fixed these citation errors in the **revised manuscript page 19, line 338, 339**.

Comment #12: Line 296: Should “C and D” be B and C?

Our response:

Thank you. We have fixed these citation errors in the **revised manuscript page 15, line 270**.

Comment #13: Lines 435-436: “To normalize the results, absorbance values from wells containing XTT solution with 436 LB media was subtracted from bacteria-containing wells.” Normalize normally means dividing. It is suggested to delete “To normalize the results”.

Our response:

Thank you for the suggestion. This phrase has been removed as you suggested.

Reviewer 2

“The authors report a platform technology that is potentially portable and high-throughput for the rapid identification of sensitive phages for clinical bacterial pathogens and thus may be helpful in personalized phage therapy. This technology was created on the basis of detecting the bioluminescence signal generated by luciferin, luciferase, and ATP in order to identify ATP molecules released during phage-mediated bacterial lysis. Additionally, they prepared a mixture of phage suspension and lyophilized ATP detection reagents for longer storage using pullulan and trehalose. Finally, they used this technology to screen phages for clinical *P. aeruginosa* isolates. Overall, the manuscript is generally well written, and easy to follow.

However, this reviewer has read the paper carefully and has concerns about the novelty of this technology and the significance of this technology for application potential in the clinic. Luminescent ATP detection has been widely used to examine active cells. Only *Pa* isolates were tested in this study.

Our response:

Thank you for taking the time to provide valuable feedback and help us increase the impact of our research.

Comment #1: This study was designed to reduce background ATP signal in phage suspensions by roughly diluting the suspension. Only 3 phages were tested on a *Pa* strain. Different phages have different features in infecting even the same host. It is not clear to this reviewer if nonlytic bacterial cells can release ATP. The background may lead to wrong conclusions.

Our response:

A. Question regarding background noise: In the revised manuscript, we have included a significantly wider range of bacterial strains and phages (**Revised manuscript, Supplementary Fig. 8, 11, and 12**). For details on the aforementioned figures please refer to response to comment 3. Our results confirm that the dilution method does in fact decrease the background ATP signal, such that the signal to noise ratio is high in the event of bacterial lysis. Phage may be purified using different techniques including PEG purification, ultrafiltration, CsCl purification, etc., which can further decrease the amount of ATP background in phage solutions.

B. Question regarding nonlytic phage: The burst of ATP detected presents itself only when the bacterial cell is compromised. In the context of phage-host interaction, that will happen either

through lytic cycle or lysis from without occurs (in cases when the MOI is very high). This is the same for a spot test (a lysis zone will appear if bacteria are killed, regardless of the mechanism) or OD₆₀₀ kill curves (triggering of lytic cycle will result in an OD decrease, regardless of whether the phage was obligate lytic or temperate, slowed growth with no lysis can also decrease the OD₆₀₀). This is why it is universally accepted that phages identified through spot test or OD₆₀₀ kill curves must go through additional characterization (*e.g.*, dilution to single plaque, sequencing to identify integrases, etc.). Please note that our method is proposed to replace culture methods (specifically spot tests) as the first step for phage screening. We do not claim to replace all the downstream characterization, which will have to be carried out with different steps depending on the application of choice and regardless of the screening method.

Comment #2: The optimal MOI and detection time point vary for different bacterial strains and phages and thus can impact the final signal detection results, suggesting that this technology may not be fit for large-scale screening of phages for various bacterial strains.

Our response:

This is correct. Variability of MOIs and detection speed between different phage-bacteria pairs complicates any method of screening, regardless of whether it is ATP detection, optical density kill curves, or spot tests. For example, depending on the initial phage and bacteria concentration, optical density-based detection of phage activity can take anything from 45 minutes to 3.5 hour for different *E. coli* phages.¹ In that sense, our method, just like others, is limited by the restriction imposed on it by biology. It is noteworthy that we found an MOI of 10 to be appropriate for most phage-bacteria pairs investigated in the manuscript. However, even at lower MOIs, a relatively strong signal was detected, though it longer times to detect ATP (cell lysis). We found this to be similar to OD kill-curves.

Comment #3: In addition to Pa isolates, more clinical bacterial strains (*e.g.*, *Klebsiella pneumoniae*, *Acinetobacter baumannii*, et al.) are needed to test on this technology platform.

Our response:

This is an excellent suggestion, which led to a very fruitful collaboration with the largest phage biobank in Canada, namely the Felix d'Herelle Reference Center for Bacterial Viruses at

University Laval (<https://www.phage.ulaval.ca/en/home/>). We have now added two additional bacterial species, namely *Salmonella enterica* and *Escherichia coli* (2 different strains each) with a significantly larger phage library of 16 and 28 additional phages (list of new bacteria and phages used can be found in **revised manuscript, supplementary Tables 2 and 3**). The new data has been added to **revised manuscript, Fig. 5 and revised manuscript Supplementary Fig. 11 and 12**, and also presented below. Please note that we also obtained data for *Staphylococcus aureus*, but since we had no time to benchmark against culture methods, we decided not to include those in the manuscript.

The kinetic data for both ATP assay and OD kill curves shows strong agreement with each other and with the spot test method for identifying the same phage targets. For the sake of comparison, data was rated for each method with a minus sign (no signal) or from one (+, weakest) to 3 plus signs (+++, strongest signal). The comparative data is presented in **revised manuscript Supplementary Tables 4 and 5**. Please see response to comment # 4 for discussion of the tables. These additional results were reflected in the abstract (**Revised manuscript, page 1, lines 15-17**) and the Introduction (**Revised manuscript, page 4, lines 85-86**). Experimental detail was added to **revised manuscript, page 31, lines 580-607**. Description and discussion were added to **revised manuscript, page 21, lines 374-388**, to read:

“To further challenge our platform technology, we screened two different strains of *E. coli* (O157: H7 and LF82) against a library of 28 *E. coli* phages and two strains of *S. enterica* (serovar Newport and serovar Senftenberg) against a library of 16 Salmonella phages from the Felix D’Herelle Reference Center for Bacterial Viruses, in the presence of sugar polymers. A full list of coded phages present in these two libraries can be found in **Supplementary Table 2 and 3**. The received phages had an titer of 10^8 - 10^{10} PFU/mL and were used without dilution to closely mimic a real life scenario where our technology is proposed to be used, when the initial titers in the library are not known for the bacteria of interest. **Fig. 5 d and e** shows a snapshot of the signals from the salmonella phage library after 90 min, with signals from identified targets clearly higher than the background signal. Examining the kinetic bioluminescence curves (**Supplementary Fig. 11**) show that the signal to noise ratio is very high at 90 min, clearly marking the phages that lead to lysis of the bacterial cell and burst release of ATP. The same trend can be observed for the *E. coli* phage library, where the kinetic bioluminescence curves (**Supplementary Fig. 12**) show that phages causing lysis of the bacterial cell and burst release of ATP are identified with a high signal to noise ratio within 100 min (3D snapshot presented in **Fig. 5 f and g**).”

Figure 5. Large-scale phage library screening of stabilized phage biobank. **a** Proposed workflow for screening a bacterial isolate against a phage library using our platform technology, where the phage library is screened by detecting phage-mediated ATP release using ATP bioluminescence assay in the presence of sugar polymers stabilizers. The assay can be conducted with fresh liquid sugar-polymer based ATP reagent solution or conducted on reconstituted all-inclusive sugar-based tablets. **b** The urinary tract infection isolate, *P. aeruginosa* C0072 and **c** the arm infection isolate, *P. aeruginosa* C0335 were screened against an in-house library of all-

inclusive tablets. Bioluminescence signal was recorded every 5 minutes and the snapshot of the signal at time = 60 min is shown for each well. **d** The sewage isolate, *S. enterica* serovar Newport and **e** the human blood isolate, *S. enterica* serovar Senftenberg were screened against a library of *Salmonella* phages containing 16 phages (listed in **Supplementary table 2**) in the presence of sugar polymer stabilizers. The presented data are the snapshots at 90 min time point for all phages. **f** The fecal isolate, *E. coli* O157: H7 and **g** The Crohn's disease isolate, *E. coli* LF82 were screened against a library of *E. coli* phages containing 28 *E. coli* phages listed in **Supplementary table 2** in the presence of sugar polymer stabilizers. The presented data are the snapshot at 100 min time point for all phages. Data points represent average of at least 3 replicates.

Supplementary Figure 11. ATP and OD₆₀₀ kinetic curves and spot test of two *Salmonella* strains against 16 *Salmonella* phages. a *Salmonella enterica* serovar Newport. b *Salmonella enterica* serovar Senftenberg. ATP one-pot assay was conducted in the presence of sugar polymers. The ATP and OD₆₀₀ were conducted in triplicates (n=3), spot tests were repeated independently twice. The error bars show standard deviation from the mean.

Supplementary Figure 12. ATP and OD₆₀₀ kinetic curves and spot test of two *E. coli* strains against 28 *E. coli* phages. a *E. coli* O157:H7 strain (human fecal isolate). b LF82 (Crohn's disease isolate¹). ATP one-pot assay was conducted in the presence of sugar polymers. The ATP and OD₆₀₀ were conducted in triplicates (n=3).

Comment #4: To my understanding, compared to this technology, the sensitive phages can be screened by simply detecting killing curves within several hours. It is not clear the efficiency of this method only if authors can compare them.

Our response:

The reviewer is correct to point out that kill curves, or optical density (OD₆₀₀) monitoring of liquid cultures of bacteria mixed with phages, will determine phage targets. We acknowledge that the kill curves, plaque assays as well as a range of colorimetric methods based on metabolic activity quantification, have potential for phage screening. Yet, many of them (kill curves, plaque assays) take significantly more time to obtain the results than our proposed ATP detection. In addition, these other assays require a significant amount of time of preparation, whereas here phages are ready to use as they are stored in a tablet format. To clarify, we have also added this description to **revised manuscript, page 3, lines 48-57**:

“In the common spot tests, as a result of phage lysing bacterial cells and the release of progeny phages, clear zones are formed on solid medium⁹. Liquid assays are also performed to monitor bacterial lysis as a result of phage infection and they include tracking the bacteria culture turbidity (optical density) as a result of phage lysis¹¹. The main hurdle of monitoring optical density of a bacteria culture is that bacteria cell debris can contribute to turbidity of the culture and affect the results negatively.¹² The overall metabolic activity of phage-infected bacterial cultures can also be monitored using various biochemical assays that have gained much attention recently and can be used to analyze properties related to phenotypes, namely cell growth and respiration.¹² However, these metabolic assays are detecting the phage-mediated bacterial cell lysis indirectly.”

We believe that the ATP detection method offers a significant edge over the other methods as it removes *a major bias* shared by optical density and metabolic activity monitoring, namely detection of slowed growth not accompanied by bacterial lysis (which happens for chronic phage or in some cases of lysogeny) vs bacterial cell lysis. The burst of ATP detected in our method can only happen if the bacterial cell lysed and thus detection of phage-mediated ATP release rapidly detects cell lysis. Moreover, ATP release shows a real time measurement of bacterial cell lysis which can provide information on phage dormant period on a particular bacterial strain.

Keeping this major advantage in mind, we have also performed OD kill curves for all the phages added to the revised version (**Revised manuscript, supplementary Fig. 8, 11, and 12**). For comparison purposes, the data was rated for each method with either a minus sign (-, no signal) or from one (+, weakest) to 5 plus signs (+++, strongest signal). The data is presented in **revised**

manuscript supplementary Tables 4 and 5. The kinetics data for both ATP assay and OD kill curves shows strong agreement with each other and with the spot test method. It is important to note that each method has its own bias, notably OD kill curves show similar degrees of growth suppression for different phages (same host) where plaque assays show a range of signal strength, including many cases of veiled/turbid plaques. We observed two *E. coli* phages with positive spot test that upon serial dilution showed no plaques which was in agreement with low or negative ATP signal, whereas OD kill curves showed bacterial growth suppression. These comparative data highlight the advantages of direct measurement of phage activity using methods like ATP detection. Discussion was added to **revised manuscript, page 21, lines 389- page 22, line 404**, to read:

“In addition, we benchmarked our method against gold standard culture methods, namely spot tests on semi solid media (end point detection) and kill curves (monitoring culture optical density at 600 nm, OD600). These data are presented in **Supplementary Fig. 11 and 12** and show strong agreement with culture methods, verifying the reliability of our platform. Examining the data summarized for comparison with bioluminescence assay in **Supplementary Tables 4 and 5** further reveals the utility of ATP detection as a direct measure of bacterial lysis, as opposed to indirect measures such as kill curves, namely that certain phages that showed a strong signal with kill curves, showed no ATP signal, and no or very faint spot test clearing. What gives the ATP detection method an edge over other components is that it relieves a major bias shared by optical density and metabolic activity monitoring, namely detection of slowed growth not accompanied by bacterial lysis (which happens for chronic phage or in some cases of lysogeny) vs bacterial cell lysis and destruction. The burst of ATP detected in our method can only happen if the bacterial cell is compromised and thus the detection of phage-mediated ATP release directly detects cell lysis, while other methods measure infectivity indirectly by monitoring the phenotypic features of bacterial cultures. Moreover, ATP release shows a real time measurement of bacterial cell lysis which can provide information on phage dormant period on a particular bacterial strain.”

Supplementary Table 4. Comparative rating of the ATP detection method with culture techniques for *Salmonella* phages

Salmonella enterica Serovar Newport			
Targets identified	ATP detection	OD kill curve	Spot test
SAL01	+++	+++	+++
SAL02	++	+++	+ (v)
SAL03	++	+++	+ (v)
SAL04	+++	+++	++ (v)
SAL06	No	+++	++ (v)
Salmonella enterica Serovar Senftenberg			
SAL01	+++	+++	+++

SAL02	++	+++	-
SAL03	+++	+++	+++ (v)
SAL08	+	+	+(v)
SAL11	+	+	+(v)
SAL13	-	+	+(v)*

+++: strong signal (ATP assay: the highest and continuous RLU signal detected upon strain infection within the phage library, OD₆₀₀: the lowest turbidity detected upon strain infection within the phage library in comparison with negative control (uninfected strain), Spot test: clear plaque on the strain's bacterial lawn)

++: medium signal (ATP assay: a medium RLU signal detected upon strain infection in comparison with the highest RLU value within the phage library, OD₆₀₀: the medium reduction in turbidity detected upon strain infection in comparison with negative control, Spot test: turbid plaque on the strain's bacterial lawn)

+: weak signal (ATP assay: a weak RLU signal value detected upon strain infection in comparison with the highest and medium RLU value within the phage library, OD₆₀₀: a weak reduction in turbidity detected upon strain infection in comparison with negative control, Spot test: faint plaque on the strain bacterial lawn)

v: veiled plaque

*: very faint footprint

Supplementary Table 5. Comparative rating of the ATP detection method with culture techniques for *E. coli* phages

E. coli O157: H7			
Targets identified	ATP detection	OD kill curve	Spot test
ECL03	+	+	+
ECL04	+	+++	+++
ECL07	+	+++	+
ECL08	+	+++	+++
ECL09	+	+++	+++
ECL10	++	+++	NA *
ECL12	+	+++	+++
ECL13	+	+++	+
ECL14	+	+++	+
ECL15	+++	+++	+++
ECL16	+++	+++	+++
ECL18	++	++	+
ECL25	+	+++	+++
ECL26	+++	+++	++
ECL27	+	+++	+++
ECL28	+++	+++	+++
E. coli LF82			
ECL04	+	+	+
ECL07	+	No	No
ECL08	+	+	+++
ECL09	++	++	+++
ECL12	+	No	No
ECL13	No	No	+ **
ECL15	+	++	++
ECL16	+	++	++
ECL17	+	+	+
ECL21	No	+++	+ ***
ECL22	No	+++	+ ***
ECL23	+	No	No
ECL24	+	No	No
ECL25	+++	+++	+++
ECL26	+	No	No
ECL27	+++	+++	+++
ECL28	+++	+++	+++

+++: strong signal (ATP assay: the highest and continuous RLU signal detected upon strain infection within the phage library, OD₆₀₀: the lowest turbidity detected upon strain infection within the phage library in comparison with negative control (uninfected strain), Spot test: clear plaque on the strain's bacterial lawn)

++: medium signal (ATP assay: a medium RLU signal detected upon strain infection in comparison with the highest RLU value within the phage library, OD₆₀₀: the medium reduction in turbidity detected upon strain infection in comparison with negative control, Spot test: turbid plaque on the strain's bacterial lawn)

+: weak signal (ATP assay: a weak RLU signal value detected upon strain infection in comparison with the highest and medium RLU value within the phage library, OD₆₀₀: a weak reduction in turbidity detected upon strain infection in comparison with negative control, Spot test: faint plaque on the strain bacterial lawn)

*: The kill curve and ATP detection methods exhibit relatively strong signals, leading to the conclusion that the spot test in this case may be biased.

** : very faint footprint

***: no plaques detected upon serial dilution.

Minor comments:

Comment #1: Lines 104-206, Podoviridae and Myoviridae have been discarded by the ICTV. Please update.

Our response:

We recognize that after the 2022 ratification vote, the morphology-based families *Myoviridae*, *Podoviridae*, and *Siphoviridae* have been abolished and the order *Caudovirales* was replaced by the class *Caudoviricetes* to group all tailed bacterial and archaeal viruses with icosahedral capsids and a double-stranded DNA genome. However, the ICTV recognises the importance of morphological (non-taxonomic) identifiers such as "podovirus", "myovirus", or "siphovirus" and accepts their use freely to reflect these distinctive features and retain their historical reference.⁴ We have therefore revised these terms to make it clear that we do not use them as taxonomic classifiers. The following information has been added to the **revised manuscript on Page 7, line 111 -112:**

“Phage *P32* is a podovirus and has a very short tail (**Fig. 1c**).³ Phage *JG004* is a myovirus phage,⁴ with an isometric head and a contractile tail (**Fig. 1c**).

Comment #2: Line 227. This hypothesis *in vitro* may not be correct in the real-life application of phage therapy *in vivo*.

Our response:

The reviewer is correct to point out that *in vitro* and *in vivo* conditions are not similar, which may affect efficacy of phage therapy. Our platform is proposed to replace culture techniques

as the *first step* in therapeutic phages identification, allowing for rapid screening of large phage libraries. The results achieved are comparable to the century old culture techniques, in that the identified phages “potential targets” only and will have to go through additional characterization and processing before being implemented as part of a therapeutic regimen. Therefore, how well a phage will perform *in vivo* is not something our platform or the competing culture techniques can address during the screening stage.

Comment #3: Line 230. This claim may be not correct.

Our response:

Thank you for bringing this to our attention. The claim in reference is that “Phages with short latent periods and large burst sizes are hypothesized to be more effective for biocontrol.⁵”. We recognize that phage burst size and latent period, even though commonly referred to as measures of virulency in the literature, are not necessarily indicative of the most effective therapeutic phages, especially *in vivo*.⁶ We have replaced the aforementioned sentence in the **revised manuscript, page 13, line 230-233**, to read:

“Latent periods and burst size of phages are among the characteristics used to evaluate phage virulence,²⁷ but they may not be indicative of therapeutic potential *in vivo* as various conditions will affect the efficacy of phage therapy as mentioned by other researchers.²⁸”

Comment #4: Figure 4D and Figure 4G were not cited. Line 276, Figure 4B was wrongly cited.

Our response:

Thanks for bringing this to our attention. The error has been fixed in the **revised manuscript, page 15, line 268 and page 17, line 320**.

Comment #5: Line 318, How to define the efficiency of plaquing the identified phage?

Our response:

That is a very important consideration in phage application, and one that will have to be addressed with established microbiology techniques after target phage have been identified. It may even lead to changing to phage production host to take into account EOPs due to R-M systems for example. Our platform is proposed to replace culture techniques for the initial phage screening, during which hundreds of phages may need to be screened and thus a high throughput, low labor technology is needed. Once a set of phages has been identified (regardless of whether the identification was done by culture methods or with our method), then additional characterization and processing may have to be performed, as demanded by the downstream application (e.g., efficiency of plaquing, genome sequencing, production at high titers, processing for removal of endotoxins, etc.). Please note that we have observed that as long as the initial phage concentration is kept the same, the RLU signal intensity can be used for qualitative comparison of the phage lytic activity. We do not, however, make any claims in the manuscript based on this observation because with large set of phages, it may not be possible to normalize all phage titers.

References:

1. Thouand, G. & Marks, R. *Bioluminescence: Fundamentals and applications in biotechnology-Volume 2. Advances in Biochemical Engineering/Biotechnology* **145**, (2014).
2. Rajnovic, D., Muñoz-Berbel, X. & Mas, J. Fast phage detection and quantification: An optical density-based approach. *PLoS One* **14**, 1–14 (2019).
3. Darfeuille-Michaud, A. *et al.* Presence of adherent *Escherichia coli* strains in ileal mucosa of patients with Crohn's disease. *Gastroenterology* **115**, 1405–1413 (1998).
4. Turner, D. *et al.* Abolishment of morphology-based taxa and change to binomial species names: 2022 taxonomy update of the ICTV bacterial viruses subcommittee. *Arch. Virol.* **168**, 1–9 (2023).
5. Karumidze, N. *et al.* Characterization of lytic *Pseudomonas aeruginosa* bacteriophages via biological properties and genomic sequences. *Appl. Microbiol. Biotechnol.* **94**, 1609–1617 (2012).
6. Garbe, J., Bunk, B., Rohde, M. & Schobert, M. Sequencing and Characterization of *Pseudomonas aeruginosa* phage JG004. *BMC Microbiol.* **11**, 1–12 (2011).
7. Montso, P. K., Mlambo, V. & Ateba, C. N. Characterization of Lytic Bacteriophages Infecting Multidrug-Resistant Shiga Toxigenic Atypical *Escherichia coli* O177 Strains Isolated From Cattle Feces. *Front. Public Heal.* **7**, 1–13 (2019).
8. Gill, J. & Hyman, P. Phage Choice, Isolation, and Preparation for Phage Therapy. *Curr. Pharm. Biotechnol.* **11**, 2–14 (2010).
9. Van Belleghem, J. D., Dąbrowska, K., Vaneechoutte, M., Barr, J. J. & Bollyky, P. L. Interactions between bacteriophage, bacteria, and the mammalian immune system. *Viruses* **11**, (2019).
10. Dąbrowska, K. & Abedon, S. T. Pharmacologically Aware Phage Therapy: Pharmacodynamic and Pharmacokinetic Obstacles to Phage Antibacterial Action in Animal and Human Bodies. *Microbiol. Mol. Biol. Rev.* **83**, (2019).

Reviewers' comments:

Reviewer #2 (Remarks to the Author):

The authors' tremendous efforts to answer the criticisms are much appreciated by this reviewer. The majority of the remarks are satisfactorily answered. They compared their approach to more conventional techniques, such as the killing curve and spot test, and the results were largely consistent. However, they did not demonstrate the method's significant advantage over the conventional approaches. Rather, it is merely a substitute approach for the conventional ones.

List of Changes Made to the Revised Manuscript:

1. Additional data was provided in new *Supplementary Fig. 4* and *Supplementary Fig. 5* to support our claim about ATP signal representing phage-mediated bacteria lysis. Experimental details and discussion were added to **Page 31 , Line 580** and **Page 16, lines 283-294**, respectively (Reviewer 1, Comment #1).
2. Additional data was provided in new *Figure 5h, and i*, and *Supplementary Fig. 13* presenting utility of our method with *Staphylococcus aureus* as a gram positive bacterial species against a library of *S. aureus* phage library provided by the Felix D'Herelle Reference Center for Bacterial Viruses. The information for the two new bacterial strains are added to *Supplementary Table 1*. All new phages are presented in new *Supplementary Table 4*. Experimental detail was added to **Page 33-34, Line 633-638** and discussion was added to **Page 21-22, Lines 386-394** (Reviewer 2, Comment #1, #3, #4 and Reviewer 1, Minor Comment #2).
3. Additional information was provided in a new *Supplementary Table 7* to benchmark our method against commonly used culture methods (Reviewer 2, Comment #2).

Reponses to reviewer's comment

The authors' tremendous efforts to answer the criticisms are much appreciated by this reviewer. The majority of the remarks are satisfactorily answered. They compared their approach to more conventional techniques, such as the killing curve and spot test, and the results were largely consistent. However, they did not demonstrate the method's significant advantage over the conventional approaches. Rather, it is merely a substitute approach for the conventional ones.

Our Response

Thank you for taking the time to review our revised manuscript. We would like to take this opportunity to explicitly outline our method's significant advantage over the conventional approaches, all of which are strongly supported by our data.

1. We have engineered an all-in-one solution for preservation, screening, and transport of phage biobanks, no comparable approach exists.

We invite the reviewer to view our work as not merely a method of monitoring phage lysis, but rather an all-in-one solution specifically designed for the unique challenges of phage biobanks, the like of which does not exist and that can *fundamentally change* the way phage biobanks are stored, screened, and transported between facilities. During the intensive five-month revision process we successfully piloted our technology at the *world's largest biobank for referenced phages* (Université Laval) and demonstrated how transformative our method could be for the curators of phage biobanks.

The state of supporting technologies for phage therapy/biocontrol is decades behind that of small molecule antimicrobials such as antibiotics. Currently phage biobanks are stored as vials of liquid at 4 °C or frozen at -80°C as well as in a dried state, namely in lyophilized form stored at 4°C, which are all costly, requiring a large footprint for storage and reliant on a cold chain. Phage biobanks are rarely transported from one lab to another, there is simply no good method of doing so and besides duplication of a phage biobank takes months if adequate numbers of trained personnel are available. Instead, bacteria (in the context of phage therapy, these are often superbugs resistant to all available antibiotics) are transported between multiple labs for phage susceptibility profiling. This is archaic, costly, and a biosecurity risk, not to mention incompatible with treatment strategies of acute infections.

Our work addresses a major gap in know-how in translational phage research and is the result of over 6 years of intensive research and a joint effort between 4 research labs across two universities. As part of our team, we have the world-renowned expert on stabilization of biologics and the curator of the *world's largest collection of referenced phages*, and some of the world's most reputable phage researchers, who deal daily with the major limitations of the conventional methods referenced by the reviewer. Our team has unmatched experience with the challenges of large phage collections and have experienced first-hand how the century old conventional methods, while adequate for dealing with a limited number of phages in a research setting, fall short when working with a larger number of phages and especially with phage biobanks. As the demand for phage biocontrol increases, the size of phage biobanks will increase, as it has over the past few years, and that requires an overhaul in the methods and protocols used to store, preserve, screen, and transport phage biobanks.

The stabilization of all assay reagents along with different phages in a dried tablet form in a microtiter plate eliminates the need lengthy reagents preparation step by providing ready to use microtiter plates that can be simply tested by rehydrating and adding the bacterial culture. It further decreases the footprint and cost for storage of phage biobanks and the cost and biosecurity risk for transporting these biobanks from the reference library to the point of need. Please keep in mind that a centralized phage biobank does not exist, and so multiple phage biobanks may need to be shipped to the point of need to save a patient's life.

2. We developed a *direct* method for real-time monitoring of phage-mediated bacterial lysis.

Conventional methods offer only an indirect view of phage activity and are thus flawed and biased.

There are currently no reports of high throughput direct and real-time monitoring of phage-mediated cell lysis in the literature. Susceptibility screening for larger phage collections is largely carried out with spot tests, not because it is the most reliable or bias free method, but because it is the most hassle-free method for screening tens or sometimes hundreds of phages with a lower chance of cross contamination.

While we acknowledge that the turbidity monitoring (kill curves), spot tests, and a range of colorimetric methods based on metabolic activity quantification can be used for phage screening, no method is free of bias. For example, the main bias of monitoring optical density of a bacteria culture during phage lysis is that bacteria cell debris can contribute to turbidity of the culture and affect the results negatively. In addition, optical density and metabolic activity monitoring can mistakenly detect slowed growth not accompanied by bacterial lysis (which happens for chronic phage or in some cases of lysogeny) as bacterial cell lysis. The burst of ATP detected in our method can only happen if the bacterial cell is compromised and thus the detection of phage-mediated ATP release *directly* detects cell lysis, while other methods measure infectivity *indirectly* by monitoring the phenotypic features of bacterial cultures.

By providing a direct and real time measurement of bacterial cell lysis through ATP release can further shed light on phage-bacteria interactions. For example, it can provide information on phage dormant period in a particular bacterial strain, compare the lytic efficiency of phages without the bias of slowed growth detection, and help better characterize phages by providing information such as latent period.

3. We stabilized ATP bioluminescence assay reagents at physiological temperature, currently only possible through genetic engineering.

ATP bioluminescence assays are usually conducted at room temperature due to the instability of enzymes at higher temperatures. Bacterial culture is sampled at specific timepoints and mixed with the ATP reagent solution. There are no existing reports of using pullulan-trehalose to stabilize ATP bioluminescence assay reagents at 37 °C. Experts in the field are aware of the limitation posed by the temperature sensitivity of these enzymes, but the existing literature is focused on genetic modification of beetle or firefly luciferase to increase thermal stability. The presence of pullulan and trehalose can preserve the ATP bioluminescence assay components at 37°C effectively and provide the opportunity to detect the phage-mediated ATP release in real time. Our method opens a host of applications for the ATP bioluminescence assay without the complexities of genetic modification.

As you can read from the above, we make a strong case that significant advances have been made in our manuscript, head and shoulders above the status quo. We have added additional evidence supporting this claim to the most recent version of the manuscript, outlined in the next pages.

Additional Changes

1. Supplementary Figures 4 and 5: To further validate the central hypotheses of our work, the activity of the ATP reagent solution with and without sugar polymers had been previously tested at the beginning of the assay, and after 1 and 3 hours. Now, we have added stability data for 6-hour heat exposure of enzymes in the presence and absence of sugars to the previous data presented in **Supplementary Fig. 4 and 5**.

Briefly, in **Supplementary Fig. 4 and 5**, we designed an experiment with ATP standard solutions as suggested. ATP reaction solutions were prepared in the absence and presence of the sugar polymer mixture. ATP standard solutions with three different concentrations (0.4, 0.01, and 0.001 μM) were prepared including in water. The 0.4 μM ATP standard solution was chosen due to RLU values at least 10 times larger than the amount of ATP released in the solution for the phage-mediated ATP detection in our one-pot assays as suggested. The other two additional concentrations of 0.01 and 0.001 μM ATP standard solutions were also tested to see if at lower ATP values we are still able to detect consistent signal in real-time phage mediated RLU signal monitoring at 37°C for weak lysis activity. The results (**revised manuscript Supplementary Fig.4**) confirmed that sugar polymer solutions were able to stabilize the ATP bioluminescence assay enzymes at 37°C, and the loss of signal was not significant even after 6 hours incubation at 37°C. On the other hand, the ATP reagent solution in the absence of sugars led to significant decrease in RLU signal after incubation at 37°C (**revised manuscript supplementary Fig. 5**). In addition, the kinetics of the RLU signal confirmed that the rate of signal loss was also higher when no sugar was present in the ATP reagent solution. Specifically, the loss of signal after 3 and 6 hours of exposure to heat at 37°C was approximately 90%. This shows that the ATP bioluminescence assay in the presence of sugar polymers is able to monitor the real-time ATP release following phage-mediated bacterial cell lysis.

The revised text in the manuscript (**Revised manuscript, page 16, lines 283-294**) reads:

“The activity of the ATP reagent solution with and without sugar polymers were also tested in four conditions including no previous exposure at 37°C (tested right after preparation at room temperature), one, three and 6 hours after incubation at 37°C. ATP standard solutions were added to the above-mentioned treated and untreated ATP reactions in the presence and absence of the sugar mixture, and the signal was measured at time zero, followed by continuous measurements for up to 3 hours. The results confirmed that sugar polymer was able to stabilize the ATP bioluminescence assay at 37°C, and the loss of signal was not significant after exposure at 37°C

in all three ATP standard concentrations. On the other hand, in the absence of sugars, the ATP reagent solution led to a significant decrease in RLU signal after incubation at 37°C (Supplementary Fig. 4). In addition, the kinetic RLU signal monitoring showed that the rate of signal loss was also higher when no sugar was present in the ATP reagent solution (Supplementary Fig. 4). Specifically, the loss of signal after 6 hours incubation at 37°C resulted in an approximately 90% signal loss (Supplementary Fig. 5). This confirms that ATP bioluminescence assay in the presence of sugar polymers has been able to detect the ATP release as a result phage-mediated bacterial cell lysis.”

Supplementary Figure 4. Effect of sugar polymers on stability of the ATP reagent solution at 37°C. ATP standard solutions were added to ATP reagent solution with (+) and without (-) sugar mixture at time =0, after incubating for one, three and six hours at 37°C, and the results were compared to no heat exposure ATP reagent solution. ATP standard solutions are used at **a** 0.4 μM, **b** 0.01 μM, and **c** 0.001 μM. The RLU signal was measured kinetically every 10 minutes for up to 3 hours. Sugar mixture was able to preserve the activity of luciferin and luciferase as there were no significant changes in the RLU signals after heat exposures. In the absence of sugars, the activity of ATP reagent solution decreased as significantly lower RLU signals were detected, which was more drastic after 3 and 6 hours of heat exposure. The RLU signal also dropped faster in the absence of sugar mixture. The presented data are the average of three replicates (n=3) with error bars representing standard deviation from the mean.

Supplementary Figure 5. Comparison of the RLU signal intensity of heat treated and untreated ATP reaction solutions in the presence and absence of sugar mixture after heat exposure at 37°C. RLU signal at time=0, right after adding ATP standard solution with 0.4 μM, 0.01 μM, 0.001 μM to ATP reagent solutions with (+) and without (-) sugar mixture after incubating for 1, 3 and 6 hours at 37°C in comparison with no heat exposure ATP reagent solution (time=0). The presented data are the average of three replicates (n=3) with Standard deviation from the mean. Statistical significance in all panels is derived from Two-way analysis of variance (ANOVA). Significance levels include *P < 0.05, **P < 0.01, ***P < 0.001, and ****P < 0.0001.

2. Additional data for *S. aureus* 68 and 44A strains screened against *S. aureus* phage library

We have included another set of data for *S. aureus*, a gram positive bacteria, to have a thorough investigation for some of the main multi-drug resistant bacterial species. Two *S. aureus* strains including 68 and 44A were screened against our library containing nine *S. aureus* phages received from the Felix d'Herelle Reference Center for Bacterial Viruses at University Laval (<https://www.phage.ulaval.ca/en/home/>). The information for the two new bacterial and phages used can be found in **revised manuscript, supplementary Tables 1 and 4**. The new data has been added to **revised manuscript, Fig. 5h and i and revised manuscript Supplementary Fig. 13**, and also presented below. The kinetic data for both ATP assay and OD kill curves shows strong agreement with each other and with the spot test method for identifying the same phage targets.

For the sake of comparison, data was rated for each method with a minus sign (no signal) or from one (+, weakest) to 3 plus signs (+++, strongest signal). The comparative data is presented in **revised manuscript Supplementary Table 7**. These additional results were reflected in the abstract (**Revised manuscript, page 1, line 17**) and the Introduction (**Revised manuscript, page 4, lines 86**). Experimental detail was added to **revised manuscript, page 33-34, lines 633-638**. Description and discussion were added to **revised manuscript, page 21-22, lines 386-394, Page 24, lines 428-430** to read:

“In the *S. aureus* phage library, five phages were identified withing 170 min against the *S. aureus* 68 strain (Fig. 5h). However, a wide range of response times observed when screening this strain against our phage library as evident in the kinetic one-pot ATP curves (**Supplementary Fig. 13a**). Among the five target phages against *S. aureus* 68 strain, STA03 and STA06 showed ATP signal within 30 min and STA04 within 60 min. On the other hand, STA05, and STA07, both with very weak spot test signal, showed bioluminescence signal at later time within 130 min and 170 min, respectively. For *S. aureus* 44A, all the target phages were identified within 100 min (Fig. 5i). The kinetic one-pot ATP curves for *S. aureus* 44A is shown in **Supplementary Fig. 13b**.”

Figure 5. Large-scale phage library screening of stabilized phage biobank. a Proposed workflow for screening a bacterial isolate against a phage library using our platform technology, where the phage library is screened by detecting phage-mediated ATP release using ATP bioluminescence assay in the presence of sugar polymers stabilizers. The assay can be conducted with fresh liquid sugar-polymer based ATP reagent solution or conducted on reconstituted all-inclusive sugar-based tablets. b The urinary tract infection isolate, *P. aeruginosa* C0072 and c the arm infection isolate, *P. aeruginosa* C0335 were screened against an in-house library of all-inclusive tablets. Bioluminescence signal was recorded every 5 minutes and the snapshot of the signal at time = 60 min is shown for each well. d The sewage isolate, *S. enterica* serovar Newport and e the human blood isolate, *S. enterica* serovar Senftenberg were screened against a library of Salmonella phages containing 16 phages (listed in Supplementary table 2) in the presence of sugar polymer stabilizers. The presented data are the snapshots at 90 min time point for both Salmonella strains. f The fecal isolate, *E. coli* O157: H7 and g The Crohn’s disease isolate, *E. coli* LF82 were screened against a library of *E. coli* phages containing 28 *E. coli* phages listed in Supplementary table 3 in the presence of sugar polymer stabilizers. The presented data are the snapshot at 100 min time point for both *E. coli* strains. h The snapshot of data at time = 170 min for *S. aureus* 68 strain and g the snapshot at time=100 min for *S. aureus* 44A strain screened against a library of *S. aureus* phages containing 9 phages (listed in supplementary table 4). Data points represent average of at least 3 replicates.

Supplementary Table 1. List of bacterial species and strains

Bacteria	Identifier	Source	Note	
P. aeruginosa	PAO1	DSM 22644	DSMZ	NA
	C0072 ²	NA	Urine*	Antibiotic resistance profile: ampicillin, amoxicillin clavulanic acid, cefazolin, cefalotin, cefixime, nitrofuratoin, tetracycline, trimethoprim sulfamethoxazole, cefoxitin, ceftriaxone
	C0335 ²	NA	Arm*	ampicillin, amoxicillin clavulanic acid, cefazolin, cefalotin, cefixime, nitrofuratoin, tetracycline, trimethoprim sulfamethoxazole, cefoxitin, ceftriaxone
Salmonella	S. Newport C487	HER1019	Sewage **	NA
	S. Senftenberg S-219/89	HER1397	Human blood **	NA
E. coli	O157:H7 B1190-1	HER1262	Feces**	NA
	LF82	NA	Ileal Mucosa of Crohn’s disease	NA

			patient ¹ , provided by Dr. Brian Coombes	
S. aureus	S. aureus 68	HER 1049	NA**	NA
	S. aureus 44A	HER 1101	NA**	NA

* Clinical isolates from the in-house library at the Michael DeGroote Institute of Infectious Disease Research

** From the Felix D’Herelle Reference Center for Bacterial Viruses

Supplementary Table 4. List of *S. aureus* phages

Phage name	Codes
HER225 3A	STA01
HER226 77	STA02
HER49 P68	STA03
HER101 44AHJD	STA04
HER528 Remus	STA05
HER474 K	STA06
HER475 812	STA07
HER239 187	STA08
HER238 71	STA09

Supplementary Table 7. Comparative rating of the ATP detection method with culture techniques for *S. aureus* phages

S. aureus 68			
Targets identified	ATP detection	OD kill curve	Spot test
STA01	No	No	+*
STA03	+++	+++	+++
STA04	++	++	++
STA05	++	++	+*
STA06	++	++	+*
STA07	++	++	+*
S. aureus 44A			
STA01	No	+	+*
STA02	No	+	+*
STA03	++	++	++
STA04	+++	++	++
STA06	++	+++	++
STA07	+++	+++	++

+++ : strong signal (ATP assay: the highest and continuous RLU signal detected upon strain infection within the phage library, OD₆₀₀: the lowest turbidity detected upon strain infection within the phage library in comparison with negative control (uninfected strain), Spot test: clear plaque on the strain's bacterial lawn)

++ : medium signal (ATP assay: a medium RLU signal detected upon strain infection in comparison with the highest RLU value within the phage library, OD₆₀₀: the medium reduction in turbidity detected upon strain infection in comparison with negative control, Spot test: turbid plaque on the strain's bacterial lawn)

+ : weak signal (ATP assay: a weak RLU signal value detected upon strain infection in comparison with the highest and medium RLU value within the phage library, OD₆₀₀: a weak reduction in turbidity detected upon strain infection in comparison with negative control, Spot test: faint plaque on the strain bacterial lawn)

* : very faint footprint

REVIEWER COMMENTS

Reviewer #4 (Remarks to the Author):

The manuscript "High throughput platform technology for rapid target identification in personalized phage therapy" describes a high throughput phage screening platform for phage therapy, claimed to be a paradigm shift from slower and labour-intensive culture techniques.

Human phage therapy is indeed most effective when implemented as a personalized therapy, which requires the isolation of the bacterium that is causing the infection and the subsequent screening against large libraries of therapeutic phages, to select those able to control the pathogenic bacterium. This is laborious, but most important it's time consuming which is one of the main obstacles in a personalized phage therapy since it can jeopardize a timely and effective treatment. Moreover, the decentralized phage banks maintained locally, requires sharing the clinical strains or phages, or both, to screen for phages able to control the infection which entails issues related to transport, biological safety, costs, and stability.

The described technology seems to be a step forward, addressing all these challenges, with significance to the field. Not only ensures a high throughput screening of phages with a turnaround time compatible with a personalized therapy but also facilitates sharing between different labs. This has been shown by the data presented in the manuscript that correlates the results from the traditional methods with the developed here, and demonstrating the advantages of the latter. Although, and considering that the major advantage (and main issue in phage therapy) is the screening time, it would be important that the authors clearly state, materialize, the time taken in each methodology (highlighting the differences) when they state the benchmarking (L395). Something that concerns me, is how the method deals with the phenomenon of lysis from without. Considering that the methodology applies phages at a high MOI, the phenomenon may occur, which may lead to a false positive screening of phages. Would be important to test this and discuss the obtained results.

Some additional minor comments:

L194 Caption on panel g is missing.

L239 The intensity of the peak should be related with the number of cells lysed and not to the burst size. The authors should clarify this.

List of Changes Made to the Revised Manuscript:

1. Caption for panel **g** was added to *Figure 2* caption (**page 12, lines 200-201**, Comment #3).
2. Revised text was added to **pages 13-14, lines 239-243** in response to Comment #4.
3. *New Supplementary Figure 14* has been added to the Supplementary Information file (**page 15**) in response to Comment #2.
4. New text added to the manuscript (**pages 23-24, lines 415-448**), along with the method section added to **pages 37-38, lines 702-719**, in response to Comment #2.
5. *New Supplementary Figure 15* has been added to the Supplementary Information file (**page 16**) in response to Comment #1.
6. New text has been added to the manuscript (**pages 26-27, lines 470-493**) in response to Comment #1.

Response to reviewer's comment

The manuscript “High throughput platform technology for rapid target identification in personalized phage therapy” describes a high throughput phage screening platform for phage therapy, claimed to be a paradigm shift from slower and labour-intensive culture techniques.

Human phage therapy is indeed most effective when implemented as a personalized therapy, which requires the isolation of the bacterium that is causing the infection and the subsequent screening against large libraries of therapeutic phages, to select those able to control the pathogenic bacterium. This is laborious, but most important it is time consuming which is one of the main obstacles in a personalized phage therapy since it can jeopardize a timely and effective treatment. Moreover, the decentralized phage banks maintained locally, requires sharing the clinical strains or phages, or both, to screen for phages able to control the infection which entails issues related to transport, biological safety, costs, and stability.

The described technology seems to be a step forward, addressing all these challenges, with significance to the field. Not only ensures a high throughput screening of phages with a turnaround time compatible with a personalized therapy but also facilitates sharing between different labs. This has been shown by the data presented in the manuscript that correlates the results from the traditional methods with the developed here and demonstrating the advantages of the latter.

Our Response:

We sincerely thank the reviewer for taking time to review our work and for providing constructive feedback on our revised manuscript. We hope to have addressed all your questions.

Comment #1: Although and considering that the major advantage (and main issue in phage therapy) is the screening time, it would be important that the authors clearly state, materialize, the time taken in each methodology (highlighting the differences) when they state the benchmarking (L395).

Our Response:

The reviewer has highlighted a very important point here. Screening time, meaning the time requirement for each methodology, varies significantly between our proposed method and the culture-based method. We have added the following paragraph to the revised manuscript to explain this concept better (**Revised manuscript pages 26-27, lines 470-493**):

“The proposed new method is an all-in-one solution that is amenable to high throughput implementation, automation, and stable to store and shipping with small footprint and without the need for a cold chain. We envision phage libraries stabilized and stored in 384-well microtiter plates along with the biochemistry of cell lysis detection, in redundancy. When needed, these microtiter plates will be shipped to the point-of-use, where the only task needed will be to rehydrate and add the target bacteria; both steps are automation-friendly and take seconds to minutes (not accounting for bacterial growth of the initial bacterial culture, which is the same for all other methods methods). This is all in contrast to conventional culture-based liquid assays (kill curves), other liquid assays (e.g. XTT assay), and semi-solid assays (spot test) for which the phage biobank is seldom replicated and shipped to point-of-need, for

susceptibility screening. Because some phages may be unstable during shipping, hosts (pathogenic) strains may also need to be shipped. At the point of screening, phages will have to be retrieved from storage (in a freezer, fridge or freeze-dried) one-by-one, either added to liquid bacterial culture (tube, microtiter plate) along with the host bacteria and any other reagents that may be needed, such as redox sensitive dyes in case of an XTT assay, or divalent cations in any phage susceptibility assay of choice. In case of a spot test, the solid medium will have to be prepared in advance, which will add to the laborious preparation process. Even an ATP assay would be cumbersome and lengthy without the proposed tablets, since all reagents will have to be freshly added in a stage-wise fashion, and the assay will have to be run at room temperature, which would significantly slow down the growth of pathogenic bacteria and phage-mediated lysis, while increasing assay time. The phage stocks stored in conventional format may lose titer from one to multiple logs; so, it is standard practice to check titer prior to screening, which will add to the required time. **Supplementary Fig. 15** illustrates the overall screening time needed for our proposed method in comparison with the spot test, turbidity measurements, for a modest size library of ~100 phages and one person dedicated to the task.”

We further designed the following illustration (added to the revised manuscript as **New Supplementary Fig. 15**) to compare the overall time needed for each methodology, namely spot test, kill curve, and our method for a modest size library of ~100 phages and one full time technician dedicated to the task.

Supplementary Figure 15. Flowchart illustrating the overall time needed for susceptibility screening with various methodologies, namely spot test, kill curve, and the new proposed method, for a library of ~100 phages and one person dedicated to the task. Created with BioRender.com

Comment #2: Something that concerns me, is how the method deals with the phenomenon of lysis from without. Considering that the methodology applies phages at a high MOI, the phenomenon may occur, which may lead to a false positive screening of phages. Would be important to test this and discuss the obtained results.

Our Response: This is an excellent point. It is indeed important to evaluate the method at different MOI's and characterize the signal related to lysis from without. We have designed a new experiment to address this question and took a deep dive into the literature on lysis from without to help interpret our results.

At high MOIs, phages can bind to the bacterial strain and lead to bacterial death through abortive infection or “lysis from without” without releasing progeny phages. Cell death without progeny phage production can potentially lead to false positive results in the phage screening process. Off-target lysis of this nature is not expected to happen for all phages; it is a phenomenon that is phage dependent and can, in theory, affect any other method of phage susceptibility screening. This has been reported across conventional culture-based techniques such as turbidity measurement (kill curves) and spot test, and testing lower phage titers, as well as certain downstream assays on the shortlisted phages (*e.g.*, efficiency of plaquing), has been recommended to avoid this bias.¹

Theoretically, at MOI of 5, 96% of bacteria within a population anticipated to undergo multiple phage absorptions.² However, lysis from without is usually reported to happen at higher MOIs (>100).^{3,4} At high MOIs, the phage tail-associated lytic enzyme can hydrolyze the cell wall, leading to quick bacterial cell lysis.⁵

We originally conducted one-pot ATP assay at different MOIs in the range of 0.001 to 10 (**Figure 3c**). To address reviewer's comment, we designed a new experiment with higher MOIs and selected three different *Pseudomonas aeruginosa* strains that were infected with phage JG004 at MOIs ranging from ~0.01 to 10,000, with the bacterial concentration kept constant at 10⁷ CFU/mL (**New Supplementary Fig. 14**). It should be noted that such high MOI (10,000 or even 1,000) is not at all common in phage assays and in the manuscript, we tested our tablets with more realistic MOI's of 1-10.

As seen in the **new Supplementary Fig. 14**, only the Pa strain (PAO1) is sensitive to phage JG004, and other strains do not show any clearance on bacterial lawns (spot test) or signal in OD₆₀₀ and ATP bioluminescence assays at any MOI. **Supplementary Fig. 14a** shows the kill curves for the phage susceptible strain at different MOIs barely exhibit any difference, with the growth being effectively hindered at every MOI and the curve being visually indiscernible for MOI's ≥ 1.

The signal from ATP bioluminescence assays, however, showed very interesting trends. The Pa strain infected with extremely high MOI of 10,000, showed a significantly lower ATP bioluminescence signal compared to lower MOIs. The peak signal increased by lowering the MOI in the range of 10,000 to 10. However, the peak signal started to diminish by lowering the MOI further, while showing larger signal at the endpoint of the assay. We hypothesize that when phage concentration is very high, there might be some abortive phage infection happening due to lysis from without. It has been reported in the literature that at high MOIs, lysis from without can lead to reduced phage counts and cell death.³ This could be the reason behind the observed decrease in bioluminescence signal at MOI > 10. The additional information packed into the bioluminescence

curves highlights the power of direct monitoring of phage-mediated bacterial lysis (as opposed to indirect monitoring with turbidity assays).

The following detailed description and discussion has been added to the **Revised manuscript, pages 23-24, lines 415-448** (experimental methods were also added to the methods section in the **Revised manuscript, pages 37, lines 702-719**):

*“At high MOIs, some phages can bind to the bacterial strain and lead to bacterial death through abortive infection without releasing progeny phages, a phenomenon known as “lysis from without”. To investigate whether cell death without progeny phage production can be a source of bias in the phage screening process, we expanded the range of MOI in our investigation. Lysis from without is reported to happen at higher MOIs (>100 phages per bacterium),^{3,4} and thus we infected three *P. aeruginosa* strains with high titers of phage JG004 at a wide range of MOIs (~10,000, 1000, 100, 1, 0.1, and 0.01). As shown in **Supplementary Fig. 14**, only the *Pa* strain was sensitive to phage JG004, and the other three strains did not show any clearance on bacterial lawns (spot test) or a decrease in turbidity in standard kill curves, or any increase in RLU signal in ATP bioluminescence assay at any of the MOI’s tested (**Supplementary Fig. 14b, c, d**). As shown in **Supplementary Fig. 14a**, in the kill curves, turbidity increased at the beginning of the assay and peak turbidity decreased with increase in MOI, which could be partially attributed to lower number of uninfected bacterial cells at high phage concentration and partially to lysis from without. For the phage sensitive *Pa* strain, the kill curves at different MOIs barely exhibit any difference, with the growth being effectively hindered at every MOI and the curve being visually indiscernible at MOI’s ≥ 1 .*

*The signal from ATP bioluminescence assays, however, showed very interesting trends, highlighting the power of direct monitoring of phage-mediated bacterial lysis (as opposed to indirect monitoring with turbidity assays). The *Pa* strain infected with an extremely high MOI of ~10,000, showed a significantly lower ATP bioluminescence signal compared to lower MOIs. The peak signal increased by lowering the MOI in the range of 10,000 to 10. However, the peak signal started to diminish by lowering the MOI further, while showing larger signal at the endpoint of the assay. It is noteworthy that lysis from without usually happens earlier than normal time of lysis (<5 min), but it can also happen at normal lysis times,³ which is in line with our observation. The time to signal that we obtained for MOIs ≥ 10 was around 30 mins and increased to 60, 70 and 90 mins for MOIs 1, 0.1, and 0.01, respectively.*

*The observed trend in bioluminescence signal (**Supplementary Fig. 14**) can be explained based on the dynamic of phage-bacteria interactions. When phage concentration is extremely high, some abortive phage infection may happen due to lysis from without. It has been reported in the literature that at high MOIs, lysis from without can lead to reduced phage counts and cell death.³ This may be the reason behind the observed decrease in bioluminescence signal at MOI > 10. At MOI < 1 the signal at the endpoint was larger as the phage population was smaller and some uninfected bacteria (theoretically 37% of the population)² could continue growing, which would lead to a higher signal at the end for MOI = 0.01.”*

a) Pa:JG004

ATP bioluminescence assay

OD₆₀₀ assay

b) C0072:JG004

ATP bioluminescence assay

OD₆₀₀ assay

c) C0335:JG004

ATP bioluminescence assay

OD₆₀₀ assay

Supplementary Figure 14. Infecting *P. aeruginosa* strains with phage JG004 phage at different MOIs. Original phage JG004 titer was $\sim 10^{11}$ PFU/mL, which was diluted and spotted on the bacteria lawns. Dilution factors are shown on spot test plates ranging from 1 to 10^{-8} ($\sim 5 \times 10^{11}$ PFU/mL to $\sim 5 \times 10^3$ PFU/mL). For the ATP and OD₆₀₀ assays, bacterial subcultures were infected with phage JG004 at MOIs of 10,000, 1000, 100, 10, 1, 0.1, 0.01. **Panel a) Pa:JG004, b) C0072:JG004, c) C0335:JG004.**

Comment #3: L194 Caption on panel g is missing.

Our Response: Thank you for bringing this to our attention. The g panel caption has been added to the Figure 2 (**lines 200-201**).

Comment #4: L239 The intensity of the peak should be related with the number of cells lysed and not to the burst size. The authors should clarify this.

Our Response: This is a fair point. We agree that peak intensity is related to the number of cells lysed as the ATP released due to phage-mediated bacterial cell lysis is measured using a bioluminescence reaction. What we tried to highlight here is that when the phage has a large burst size, more progeny phages will be released into the surrounding medium after each round of phage lytic cycle. This can indirectly increase the peak intensity as more phages are available to infect the surrounding bacterial cells. In response to comment 2 and considering new experimental data, we observed that RLU signal in ATP bioluminescence assay can depend on the MOI, with very high MOIs having a reverse effect on the signal possibly due to lysis from without and lack of production of progeny phages.

To address the reviewers concern, we have revised the manuscript text to explain this clearly (**Revised manuscript, pages 13-14, lines 239-243**):

“The fact that phage P32 generated a stronger peak signal as compared to phage JG004 showed that more bacterial cells were lysed by P32 compared to JG004 (Fig. 3d). The peak signal correlates with the number of bacterial cells lysed. Having a large burst size can also affect the peak signal by having more progeny phages released to the surrounding medium, that can in turn infect and lyse more bacterial cells.”

References

1. Kutter, E. Phage Host Range and Efficiency of Plating. in *Bacteriophages. Methods in Molecular Biology*TM 141–149 (Humana Press, 2009). doi:10.1007/978-1-60327-164-6_14
2. Abedon, S. T. & Katsaounis, T. I. Basic Phage Mathematics. in *Bacteriophages. Methods in Molecular Biology* 3–30 (Springer, 2018). doi:10.1007/978-1-4939-7343-9_1
3. Abedon, S. T. Lysis from without. *Bacteriophage* **1**, 46–49 (2011).
4. Dennehy, J. J. & Abedon, S. T. Phage Infection and Lysis. in *Bacteriophages* 1–43 (Springer, 2020). doi:10.1007/978-3-319-40598-8_53-1
5. Brown, C. M. & Bidle, K. D. Attenuation of virus production at high multiplicities of infection in *Aureococcus anophagefferens*. *Virology* **466–467**, 71–81 (2014).

REVIEWERS' COMMENTS

Reviewer #4 (Remarks to the Author):

The authors of the manuscript "High throughput platform technology for rapid target identification in personalized phage therapy" have replied to the raised questions. Still, some issues remain in the revised manuscript:

- My previous Comment #1: The authors have described the differences in the methodologies, but the time needed for each methodology is only materialized in Supplementary Fig. 15, which is not part of the main manuscript. At least one sentence comparing the turnaround times should be added to the main manuscript.

Considering Supplementary Fig. 15, the spot test does not require a specific phage titer (even very low PFUs are enough) and thus checking titer prior to screening is not required. Consequently, this step should be "minutes to hours" instead of "days to weeks".

- My previous Comment #2: The authors have put some effort to address this issue. Even so:

i) Please be aware that "abortive infection" is a phenomenon different from "Lysis from without". "Lysis from without" is when cell mortality results from numerous punctures rather than from ensuing infection and happens at high MOIs while "abortive infection" usually results from a bacterial defence mechanism after phage genome injection, not necessarily involving high MOIs. Considering this, the sentence in L415-416 and in L442-L443 should be rewritten.

ii) Although "Lysis from without" can occur in bacteria that the phage infects, the issue here would be in bacteria that the phage is not able to infect (produce phage progeny at low MOI). It would be interesting and important to find such strains to assess the effect caused in the methodology, since it can lead to a false positive screening of a phage that does not infect the target bacteria and that will not be efficient for therapy (the aim of the method). If not possible to address this, it should be stressed (reinforced) the minimum bacterial concentration to be tested in the method to avoid high MOI that can lead to this phenomenon.

Response to reviewer's comment

The authors of the manuscript “High throughput platform technology for rapid target identification in personalized phage therapy” have replied to the raised questions. Still, some issues remain in the revised manuscript:

Comment #1:

My previous Comment #1: The authors have described the differences in the methodologies, but the time needed for each methodology is only materialized in Supplementary Fig. 15, which is not part of the main manuscript. At least one sentence comparing the turnaround times should be added to the main manuscript.

Our Response: Thanks for your suggestion. We have added text to the manuscript to compare the turnaround times for each phage screening methodology. The sentence reads as follows (page 23, lines 445-449):

“The turnaround time for the all-inclusive tablets could reduce to ~2 hours, while other methods usually require a few days. By having shelf-stable phage tablets, the only step required is to reconstitute the dried tablets, add bacteria and monitor the bioluminescence signal to directly detect phage-mediated cell lysis through ATP release.”

Comment #2:

Considering Supplementary Fig. 15, the spot test does not require a specific phage titer (even very low PFUs are enough) and thus checking titer prior to screening is not required. Consequently, this step should be “minutes to hours” instead of “days to weeks”.

Our Response: This is a fair suggestion and one which we have taken into account in the revised version of the manuscript.

We included the phage titer check in Figure S15 as a means of quality control. We acknowledge that while all curators have a means for quality control during screening, the specific protocol depends on locally established best practices, which will vary somewhat from one lab to the other. Upon consultation with our coauthors who have hands-on experience with large phage biobanks, we learned that, depending on the size of the phage collection to be screened, some level of quality control using the original bacterial host will always be performed prior to or in parallel with spot test. We agree that if this were to be done not in the form of rigorous titer check, but rather a spot test on the original host in parallel to spot test on the new bacterial strain, one could save significant amount of time.

However, if during the quality control (in this case spot test on the original host in parallel to spot tests on the new bacterial strain) we find some phages to have been inactivated or have suffered significant titre loss, we will have to go back to master stocks and repropagate. Depending on the number of phages, this step could take anywhere between one to several days. Please note that in our experience, certain phages are more sensitive to storage conditions and lose titer rapidly in the absence of a deliberate preservation mechanism, which is not standard practice in phage biobanks.

We have revised the Supplementary Fig. 15 based on your comment.

Supplementary Figure 15. Flowchart illustrating the overall time needed for susceptibility screening with various methodologies, namely spot test, kill curve, and the new proposed method, for a library of ~100 phages and one person dedicated to the task.

Comment #2: My previous Comment #2: The authors have put some effort to address this issue. Even so:

i) Please be aware that “abortive infection” is a phenomenon different from “Lysis from without”. “Lysis from without” is when cell mortality results from numerous punctures rather than from ensuing infection and happens at high MOIs while “abortive infection” usually results from a bacterial defence mechanism after phage genome injection, not necessarily involving high MOIs. Considering this, the sentence in lines and in L442-L443 should be rewritten.

ii) Although “Lysis from without” can occur in bacteria that the phage infects, the issue here would be in bacteria that the phage is not able to infect (produce phage progeny at low MOI). It would be interesting and important to find such strains to assess the effect caused in the methodology, since it can lead to a false positive screening of a phage that does not infect the target bacteria and that will not be efficient for therapy (the aim of the method). If not possible to address this, it should be stressed (reinforced) the minimum bacterial concentration to be tested in the method to avoid high MOI that can lead to this phenomenon.

Our Response:

(i) Thank you very much for this comment. We acknowledge that abortive infection due to bacterial defence mechanisms after phage infection does not necessarily happen as a result of high MOI. We have revised the text mentioned in the revised manuscript, this text was added to page 20, lines 371-379:

“It is important to note that certain phenomena may result in effects that could be mistaken for the effects resulting from the completion of the lytic cycle. For example, at high MOIs (usually >100 phages per bacterium), some phages can bind to the bacterial strain and lead to cell death through abortive infection without releasing progeny phages, a phenomenon known as “lysis-from-without”.^{35,36} In addition to lysis caused by multiple phage absorption to the surface of bacterial cells, there are reports of abortive phage infection due to bacterial defence mechanism, which is not necessarily as a result of high MOI. Abortive infections can prevent phage propagation by putting bacteria in a dormant state or lead to their death, or mutual destruction of phage and bacteria.³⁷

(ii) Regarding “lysis from without” in bacteria with a non-target phage infection, we would like to point out that the results in Figure S14 do contain off-target bacteria. In addition, we have more data with off-target bacteria from other projects (not included in the manuscript) that follow the same trends reported in Figure S14. We acknowledge that off-target action is a possibility at high MOIs. If it were to happen, this phenomenon would manifest itself as a false positive in both traditional culture methods (spot test, kill curves) and direct lysis monitoring methods, such as our method of ATP release detection. What we have observed is that with methods of direct lysis monitoring, including but not limited to ATP detection, since the signal is not amplified through the production of progeny phage, it is much easier to separate signal from noise, or to put this in other words, separate false positive from a true hit. This is not to say that our method is free of bias; we don’t believe that to be true for any method. Therefore, we have added additional text to highlight that our proposed method, as with culture methods, may be affected by false positives, the rate of which we cannot quantify or compare. This text was added to the revised manuscript, page 20, lines 371-391:

“It is important to note that certain phenomena may result in effects that could be mistaken for the effects resulting from the completion of the lytic cycle. For example, at high MOIs (usually >100 phages per bacterium), some phages can bind to the bacterial strain and lead to cell death through abortive infection without releasing progeny phages, a phenomenon known as “lysis-from-without”.^{35,36} In addition to lysis caused by multiple phage absorption to the surface of bacterial cells, there are reports of abortive phage infection due to bacterial defence mechanism, which is not necessarily as a result of high MOI. Abortive infections can prevent phage propagation by putting bacteria in a dormant state or lead to their death, or mutual destruction of phage and bacteria.³⁷

Such effects can result in a zone of clearing in a spot test assay that would be indiscernible from the zone of clearing caused by the completion of the lytic cycle. However, when followed up by downstream quality control with serial dilutions, the same phage would fail to lead any plaques. In a kill curve assay that relies on turbidity monitoring, phage-related bacterial cell dormancy can show a decrease in turbidity that

can easily be mistaken for the effect of phage lysis. While methods of direct lysis monitoring, such as ATP detection, have advantages over indirect methods (*e.g.*, spot test and kill curves), they may also be affected by complex phenomenon known to exist in bacteria-phage interaction, such as lysis-from-without or abortive infection. Thus, we emphasize that our method does not replace a rigorous downstream quality control on the shortlisted phage identified through our rapid screening method, or any other method of choice. Furthermore, while the true MOI cannot be determined a priori during phage screening, theory suggests using a higher bacterial concentration may help avoid lysis-from-without.”

New added reference:

37. Aframian, N. & Eldar, A. Abortive infection antiphage defense systems: separating mechanism and phenotype. *Trends Microbiol.* **31**, 1003–1012 (2023).

Another point to note is that our method is the very first step in phage preparation for therapy/biocontrol applications. While not standard practice in all labs, it is reasonable to expect a certain level of quality control upstream and downstream of the initial biobank screening. We do not claim to have replaced the essential quality control steps that should be done for phages in a biobank (such as sequencing and screening for genes of interest) or those that currently follow the initial screening, such as efficiency of plaquing (a method that is very effective at weeding out false positives) and mutation frequency. While the field would benefit from high throughput and automated implementation of the downstream quality control steps, we believe that the initial screening is a major bottle neck in terms of timely administration of phage therapy, since a much larger number of phages are subject to screening. Whereas the downstream quality control (that will separate false positives) will be done on a subset of phages. If recent history of cases published is any indication, this short list is more likely to be a few phages and easily handled by culture methods.